# ORACLE EFFICIENT ALGORITHMS FOR GROUPWISE REGRET

**Krishna Acharya**[1]**, Eshwar Ram Arunachaleswaran**[2]**, Sampath Kannan**[2]**, Aaron Roth**[2]**, Juba Ziani**[1]
[1]Georgia Institute of Technology, [2]University of Pennsylvania
[1]{kacharya33, jziani3}@gatech.edu, [2]{eshwar,kannan,aaroth}@seas.upenn.edu

## ABSTRACT

We study the problem of online prediction, in which at each time step $t \in \{1, 2, \cdots, T\}$, an individual $x_t$ arrives, whose label we must predict. Each individual is associated with various groups, defined based on their features such as age, sex, race etc., which may intersect. Our goal is to make predictions that have regret guarantees not just overall but also simultaneously on each sub-sequence comprised of the members of any single group. Previous work (Blum & Lykouris, 2020) provides attractive regret guarantees for these problems; however, these are computationally intractable on large model classes (e.g., the set of all linear models, as used in linear regression). We show that a simple modification of the sleeping-experts-based approach of Blum & Lykouris (2020) yields an efficient *reduction* to the well-understood problem of obtaining diminishing external regret *absent group considerations*. Our approach gives similar regret guarantees compared to Blum & Lykouris (2020); however, we run in time linear in the number of groups, and are oracle-efficient in the hypothesis class. This in particular implies that our algorithm is efficient whenever the number of groups is polynomially bounded and the external-regret problem can be solved efficiently, an improvement on Blum & Lykouris (2020)'s stronger condition that the model class must be small. Our approach can handle online linear regression and online combinatorial optimization problems like online shortest paths. Beyond providing theoretical regret bounds, we evaluate this algorithm with an extensive set of experiments on synthetic data and on two real data sets — Medical costs and the Adult income dataset, both instantiated with intersecting groups defined in terms of race, sex, and other demographic characteristics. We find that uniformly across groups, our algorithm gives substantial error improvements compared to running a standard online linear regression algorithm with no groupwise regret guarantees.

## 1 INTRODUCTION

Consider the problem of predicting future healthcare costs for a population of people that is arriving dynamically and changing over time. To handle the adaptively changing nature of the problem, we might deploy an online learning algorithm that has worst-case regret guarantees for arbitrary sequences. For example, we could run the online linear regression algorithm of Azoury & Warmuth (2001), which would guarantee that the cumulative squared error of our predictions is at most (up to low-order terms) the squared error of the best fixed linear regression function in hindsight. Here we limit ourselves to a simple hypothesis class like linear regression models, because in general, efficient no-regret algorithms are not known to exist for substantially more complex hypothesis classes. It is likely however that the underlying population is heterogenous: not all sub-populations are best modeled by the same linear function. For example, "the best" linear regression model in hindsight might be different for different subsets of the population as broken out by sex, age, race, occupation, or other demographic characteristics. Yet because these demographic groups overlap — an individual belongs to some group based on race, another based on sex, and still another based on age, etc — we cannot simply run a different algorithm for each demographic group. This motives the notion of groupwise regret, first studied by Blum & Lykouris (2020). A prediction algorithm guarantees diminishing groupwise regret with respect to a set of groups $G$ and a hypothesis class $H$, if simultaneously for every group in $G$, on the subsequence of rounds consisting of individuals who are members of that group, the squared error is at most (up to low-order regret terms) the squared error of the best model in $H$ *on that subsequence*. Blum & Lykouris (2020) gave an algorithm for

solving this problem for arbitrary collections of groups $G$ and hypothesis classes $H$ — see Lee et al. (2022) for an alternative derivation of such an algorithm. Both of these algorithms have running times that depend polynomially on $|G|$ and $|H|$ – for example, the algorithm given in Blum & Lykouris (2020) is a reduction to a sleeping experts problem in which there is one expert for every pairing of groups in $G$ with hypotheses in $H$. Hence, neither algorithm would be feasible to run for the set $H$ consisting of e.g. all linear functions, which is continuously large — and which is exponentially large in the dimension even under any reasonable discretization.

Our first result is an "oracle efficient" algorithm for obtaining diminishing groupwise regret[1] for any polynomially large set of groups $G$ and any hypothesis class $H$: In other words, it is an efficient reduction from the problem of obtaining *groupwise* regret over $G$ and $H$ to the easier problem of obtaining diminishing (external) regret to the best model in $H$ ignoring group structure. This turns out to be a simple modification of the sleeping experts construction of Blum & Lykouris (2020). Because there are efficient, practical algorithms for online linear regression, a consequence of this is that we get the first efficient, practical algorithm for online linear regression for obtaining *groupwise* regret for any reasonably sized collection of groups $G$ (our algorithm needs to enumerate over the groups at each round and so has running time linear in $|G|$).

We can instantiate our algorithm for *groupwise* regret in any setting in which we have an efficient no (external) regret learning algorithm. For example, we can instantiate it for online classification problems when the benchmark class has a small separator set, using the algorithm of Syrgkanis et al. (2016); for online linear optimization problems, we can instantiate it using the "Follow the Perturbed Leader" (FTPL) algorithm of Kalai & Vempala (2005). For online linear regression problems, we can instantiate it with the online regression algorithm of Azoury & Warmuth (2001).

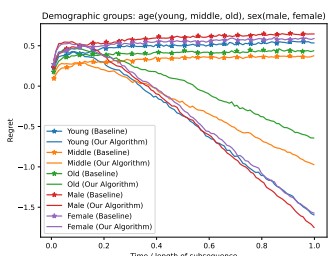

(a) Groups based on age, sex

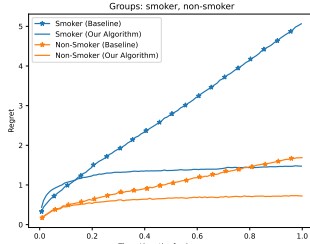

(b) Groups based on smoking habits

Figure 1: Regret vs. time (as a fraction of subsequence length), for 5 demographic groups based on age, sex in 1a, and 2 non-demographic groups based on smoking habits in 1b

With our algorithm in hand, we embark on an extensive set of experiments in which the learning task is linear regression, both on synthetic data and on two real datasets. In the synthetic data experiment, we construct a distribution with different intersecting groups. Each group is associated with a (different) underlying linear model; individuals who belong to multiple groups have labels that are generated via an aggregation function using the models of all of the containing groups. We experiment with different aggregation functions (such as using the mean, minimum, maximum generated label, as well as the label of the first relevant group in a fixed lexicographic ordering). We find that consistently across all aggregation rules, our algorithm for efficiently obtaining groupwise regret when instantiated with the online regression algorithm of Azoury & Warmuth (2001) has substantially lower cumulative loss and regret on each group compared to when we run the online regression algorithm of Azoury & Warmuth (2001) alone, on the entire sequence of examples. In fact, when using our algorithm, the regret to the best linear model in hindsight when restricted to a fixed group is often *negative*, which means we make predictions that are more accurate than that of the best linear model tailored for that group, despite using linear learners as our underlying hypothesis class. This occurs because there are many individuals who are members of multiple groups, and our algorithm must guarantee low regret on each group separately, which requires ensembling different linear models across groups whenever they intersect. This ensembling leads to an accuracy improvement.

We then run comparisons on two real datasets: the prediction task in the first is to predict patient medical costs, and the task in the second is to predict income from demographic attributes recorded in Census data. We define intersecting groups using attributes like age, sex, race etc. Here too, our algorithm, instantiated with the online regression algorithm of Azoury & Warmuth (2001), has consistently lower groupwise regret than the baseline of just running the algorithm of Azoury & Warmuth (2001) alone on the entire sequence of examples. Thus, even though algorithms with worst-

---

[1] I.e., guaranteeing sub-linear groupwise regret in the number of time steps, implying that the time-average regret vanishes and goes to 0

case regret guarantees are not known for hypothesis classes substantially beyond linear regression, we can use existing online linear regression algorithms to efficiently obtain substantially lower error while satisfying natural fairness constraints by demanding groupwise regret guarantees.

In Figure 1, we plot the performance of our algorithm instantiated with the online linear regression algorithm of Azoury & Warmuth (2001), compared to the baseline of running Azoury & Warmuth (2001) alone on all data points, on a medical cost prediction task. We divide the population into a number of intersecting groups, on the basis of age (young, middle, and old), sex (male and female) and smoking status (smoker and non-smoker). We plot the regret (the difference between the squared error obtained, and the squared error obtainable by the best linear model in hindsight) of our algorithm compared to the baseline on each of these demographic groups (in the top panel for age and sex groups, and in the bottom panel for smoking related groups). Since the number of individuals within each group is different for different groups, we normalize the $x$ axis to represent the *fraction* of individuals within each group seen so far, which allows us to plot all of the regret curves on the same scale. In these computations, the performance of our algorithm comes from a single run though the data, whereas "the best model in hindsight" is computed separately for each group. The plot records average performance over 10 runs. Here medical costs have been scaled to lie in $[0, 1]$ and so the absolute numbers are not meaningful; it is relative comparisons that are important. What we see is representative on all of our experiments: the groupwise regret across different groups is consistently both lower and growing with time at a lower rate than the regret of the baseline. The regret can even be increasingly negative as seen in the age and sex groups.

## 1.1 Related Work

The problem of obtaining diminishing groupwise regret in a sequential decision-making setting was introduced by Blum & Lykouris (2020) (see also Rothblum & Yona (2021) who introduce a related problem in the batch setting). The algorithm of Blum & Lykouris (2020) is a reduction to the sleeping experts problem; Lee et al. (2022) derive another algorithm for the same problem from first principles as part of a general online multi-objective optimization framework. Both of these algorithms have running time that scales at least linearly with (the product of) the number of groups and the cardinality of the benchmark hypothesis class. In the batch setting, Tosh & Hsu (2022) use an online-to-batch reduction together with the sleeping-experts style algorithm of Blum & Lykouris (2020) to obtain state-of-the-art batch sample complexity bounds. None of these algorithms are oracle efficient, and as far as we are aware, none of them have been implemented — in fact, Tosh & Hsu (2022) explicitly list giving an efficient version of Blum & Lykouris (2020) that avoids enumeration of the hypothesis class as an open problem. Globus-Harris et al. (2022) derive an algorithm for obtaining group-wise optimal guarantees that can be made to be "oracle efficient" by reduction to a ternary classification problem, and they give an experimental evaluation in the batch setting—but their algorithm does not work in the sequential prediction setting.

Multi-group regret guarantees are part of the "multi-group" fairness literature, which originates with Kearns et al. (2018) and Hébert-Johnson et al. (2018). In particular, multicalibration Hébert-Johnson et al. (2018) is related to simultaneously minimizing prediction loss for a benchmark class, on subsets of the data Gopalan et al. (2022; 2023); Globus-Harris et al. (2023). One way to get groupwise regret guarantees with respect to a set of groups $G$ and a benchmark class $H$ would be to promise "multicalibration" with respect to the class of functions $G \times H$. Very recently, Garg et al. (2023) gave an oracle efficient algorithm for obtaining multicalibration in the sequential prediction setting by reduction to an algorithm for obtaining diminishing external regret guarantees. However, to apply this algorithm to our problem, we would need an algorithm that promises external regret guarantees over the functions in the class $G \times H$. We do not have such an algorithm: For example the algorithm of Azoury & Warmuth (2001) can be used to efficiently obtain diminishing regret bounds with respect to squared error and the class of linear functions: but even when $H$ is the set of linear functions, taking the Cartesian product of $H$ with a set of group indicator functions will result in a class of highly non-linear functions for which online regret bounds are not known. In contrast to Garg et al. (2023), our approach can be used to give groupwise regret using a learning algorithm only for $H$ (rather than an algorithm for $G \times H$), which lets us give an efficient algorithm for an arbitrary polynomial collection of groups, and the benchmark class of linear functions.

## 2 Model

We study a model of online contextual prediction against an adversary in which we compare to benchmarks simultaneously across multiple subsequences of time steps in $\{1, \ldots, T\}$. Each subsequence

is defined by a time selection function $I : \{1, \ldots, T\} \rightarrow \{0,1\}$ [2] which specifies whether or not each round $t$ is a member of the subsequence or not. In each round, the learner observes a context $x_t \in X$ where $X \subseteq [0,1]^d$; based on this context, he chooses a prediction $p_t \in \mathcal{A} \subseteq [0,1]^n$. Then, the learner observes a cost (chosen by an adversary) for each possible prediction in $\mathcal{A}$. The learner is aware of a benchmark class of policies $H$; each $f \in H$ is a function from $X$ to $\mathcal{A}$ and maps contexts to predictions. The learner's goal is to achieve loss that is as low as the loss of the best policy in hindsight — not just overall, but also simultaneously on each of the subsequences (where "the best policy in hindsight" may be different on different subsequences).

More precisely, the interaction is as follows. The loss function $\ell$ maps actions of the learner and the adversary to a real valued loss in $[0,1]$. The time selection functions $\mathcal{I} = \{I_1, I_2, \cdots I_{|\mathcal{I}|}\}$ are the indicator function for the corresponding subsequence. In rounds $t \in \{1, \ldots, T\}$:

1. The adversary reveals context $x_t \in X$ (representing any features relevant to the prediction task at hand), as well as $I(t)$ for each $I \in \mathcal{I}$, the indicator for whether each subsequence contains round $t$.

2. The learner (with randomization) picks a policy $p_t : X \rightarrow \mathcal{A}$ [3] and makes prediction $\mathbf{p}_t(x_t) \in \mathcal{A}$. Simultaneously, the adversary selects an action $y_t$ taken from a known set $\mathcal{Y}$.

3. The learner learns the adversary's action $y_t$ and obtains loss $\ell(p_t(x_t), y_t) \in [0,1]$.

We define the regret on any subsequence *against a policy $f \in H$* as the difference in performance of the algorithm and that of using $f$ in hindsight to make predictions. The regret of the algorithm on subsequence $I$ is measured against the best benchmark policy in hindsight (for that subsequence):

$$\text{Regret on Subsequence } I = \mathbb{E}\left[\sum_t I(t)\ell(\mathbf{p}_t(x_t), y_t)\right] - \min_{f_I \in H} \sum_t I(t)\ell\left(f_I(x_t), y_t\right).$$

Here, $\mathbb{E}\left[\sum_t I(t)\ell(\mathbf{p}_t(x_t), y_t)\right]$ is the expected loss obtained on subsequence $I$ by an algorithm that uses policy $p_t$ at time step $t$, and $\min_{f_I \in H} \sum_t I(t)\ell\left(f(x_t), y_t\right)$ represents the smallest loss achievable in hindsight on this subsequence given a fixed policy $f \in H$. This regret measures, on the current subsequence, how well our algorithm is doing compared to what could have been achieved had we known the contexts $x_t$ and the actions $y_t$ in advance—if we could have made predictions for this subsequence in isolation. Of course the same example can belong to multiple subsequences, which complicates the problem. The goal is to upper bound the regret on each subsequence $I$ by some sublinear function $o(T_I)$ where $T_I = \sum_t I(t)$ is the 'length' of $I$.

In the context of our original motivation, the subsequences map to the groups we care about—the subsequence for a group is active on some round $t$ if the round $t$ individual is a member of that group. Our benchmark class can be, e.g., the set of all linear regression functions. Obtaining low subsequence regret would be straightforward if the subsequences were disjoint[4]. The main challenge is that in our setting, the subsequences can intersect, and we cannot treat them independently anymore.

In aid of our techniques, we briefly describe the problem of external regret minimization, a problem that is generalized by the setup above, but which consequently admits stronger results (better bounds in running time/ regret) in various settings.

**External Regret Minimization** In this problem, the learner has access to a finite set of experts $E$, and picks an expert $\boldsymbol{e}_t \in E$ with randomization in each round $t \in \{1, 2 \cdots T\}$, in each round. An adversary then reveals a loss vector $\ell_t \in \mathbb{R}^{|E|}$ with the learner picking up a loss of $\ell_t(e_t)$. The learner's objective is to minimize the overall regret, called external regret, on the entire sequence as measured against the single best expert in hindsight. Formally :

$$\text{External Regret} = \mathbb{E}\left[\sum_t \ell_t(\boldsymbol{e}_t)\right] - \min_{e \in E} \sum_t \ell_t(e).$$

An online learning algorithm is called a no-regret algorithm if its external regret is sublinear in $T$, i.e. $o(T)$, and its running time and regret are measured in terms of the complexity of the expert set $E$.

---

[2]Our results hold without significant modification for fractional time selection, i.e., $I : 1 \ldots T \rightarrow [0,1]$.

[3]For most of our applications, the policy is in fact picked from the benchmark class, however in one of our applications, online least regression, the analysis is simplified by allowing policies beyond the benchmark.

[4]One could just run a separate no-regret learning algorithm on all subsequences separately.

## 3 SUBSEQUENCE REGRET MINIMIZATION VIA DECOMPOSITION

We outline our main algorithm for online subsequence regret minimization. Algorithm 1 maintains $|\mathcal{I}|$ experts, each corresponding to a single subsequence, and aggregates their predictions suitably at each round. Each expert runs their own external regret minimizing or no-regret algorithms (each measuring regret against the concept class $H$). However and as mentioned above, the challenge is that any given time step might belong simultaneously to multiple subsequences. Therefore, it is not clear how to aggregate the different suggested policies corresponding to the several currently "active" subsequences (or experts) to come up with a prediction. To aggregate the policies suggested by each expert or subsequence, we use the AdaNormalHedge algorithm of Luo & Schapire (2015) with the above no-regret algorithms forming the set of $k$ meta-experts for this problem.

---

**Parameters:** Time Horizon T

1. Initialize an instance $\mathcal{Z}_I$ of an external regret minimization algorithm for the subsequence corresponding to each time selection function $I \in \mathcal{I}$, with the policy class $H$ as the set of experts.

2. Initialize an instance $\mathcal{Z}$ of AdaNormalHedge with $\{\mathcal{Z}_I\}_{I \in \mathcal{I}}$ as the set of policies, which we call meta-experts.

3. For $t = 1, 2, \cdots T$:

   (a) **Subsequence-level prediction step:** Observe context $x_t$. Each meta-expert $\mathcal{Z}_I$ recommends a randomized policy $\boldsymbol{p}_t^I$ with realization $z_t^I$ (from $H$).

   (b) **Prediction aggregation step:** Using the information from the subsequence indicators and fresh randomness, use AdaNormalHedge to sample a meta-expert $\mathcal{Z}_{I'}$ [a]. Set the policy $\boldsymbol{p}_t$ of the algorithm to be policy $z_{I'}^t$ offered by this meta-expert (i.e. pick action $z_{I'}^t(x_t)$).

   (c) **Update step:** Observe the adversary's action $y_t$. Update the state of algorithm $\mathcal{Z}$ by treating the loss of meta-expert $\mathcal{Z}_I$ as $\ell(z_t^I(x_t), y_t)$ For each subsequence $I$ that is 'active' in this round (i.e., with $I(t) = 1$), update the internal state of the algorithm $\mathcal{Z}_I$ using $x_t$ and $y_t$.

   ---
   [a]AdaNormalHedge in fact samples from only the set of meta-experts corresponding to active subsequences

Algorithm 1: Algorithm for Subsequence Regret Minimization

---

**Theorem 1.** *For each subsequence $I$, assume there is an online learning algorithm $\mathcal{Z}_I$ that picks amongst policies in $H$ mapping contexts to actions and has external regret (against $H$) upper bounded by $\alpha_I$. Then, there exists an online learning algorithm (Algorithm 1), that given $|\mathcal{I}|$ subsequences $\mathcal{I}$, has total runtime (on a per round basis) equal to the sum of runtimes of the above algorithms and a term that is a polynomial in $|\mathcal{I}|$ and $d$ and obtains a regret of $\alpha_I + O\left(\sqrt{T_I \log |\mathcal{I}|}\right)$ for any subsequence $I$ with associated length $T_I$.*

In the interest of space, the proof has been moved to Appendix C. As an immediate consequence of this theorem, we obtain the main result of our paper as a corollary (informally stated).

**Theorem 2.** *Any online learning problem with a computationally efficient algorithm for obtaining vanishing regret also admits a computationally efficient algorithm for vanishing subsequence regret over subsequences defined by polynomially many time selection functions.*

**Improved computational efficiency compared to Blum & Lykouris (2020)** We remark that our method is similar to Blum & Lykouris (2020) in that it relies on using AdaNormalHedge to decide between different policies. However, it differs in the set of actions or experts available to this algorithm, allowing us to obtain better computational efficiency in many settings.

In our algorithm, we use exactly $|\mathcal{I}|$ experts; each of these experts corresponds to a single subsequence and makes recommendations based on a regret minimization algorithm run only on that subsequence. Their construction instead instantiates an expert for each tuple $(f, I)$, where $f \in H$ is a policy and $I \in \mathcal{I}$ is a subsequence; a consequence is that their running time is polynomial in the size of the hypothesis class. We avoid this issue by delegating the question of dealing with the complexity of the policy class to the external regret minimization algorithm associated with each subsequence, and by only enumerating $|\mathcal{I}|$ experts (one for each subsequence). Practical algorithms are known for the

external regret minimization sub-problem (that must be solved by each expert) for many settings of interest with large (or infinite) but structured policy classes (expanded upon in Section 4).

## 4 APPLICATIONS OF OUR FRAMEWORK

### 4.1 ONLINE LINEAR REGRESSION

Our first application is online linear regression with multiple groups. At each round $t$, a single individual arrives with context $x_t$ and belongs to several groups; the learner observes this context and group membership for each group and must predict a label.

Formally, the context domain is $X = [0, 1]^d$ where a context is an individual's feature vector; the learner makes a prediction $\hat{y}_t$ in $\mathcal{A} = [0, 1]$, the predicted label ; the adversary, for its action, picks $y_t \in \mathcal{Y} = [0, 1]$ as his action, which is the true label; and the policy class $H$ is the set of linear regressors in $d$ dimensions, i.e. $H$ consists of all policies $\theta \in \mathbb{R}^d$ where $\theta(x_t) := \langle \theta, x_t \rangle \; \forall x_t \in X$.[5] Each subsequence $I$ corresponds to a single group $g$ within the set of groups $G = \{g_1, g_2, \cdots g_{|G|}\}$; in each round the subsequence indicator functions $I(t)$ are substituted by group membership indicator functions $g(t)$, for each $g \in G$. The goal is to minimize regret (as defined in Section 2) with respect to the squared error loss: i.e., given that $y_t$ is the true label at round $t$, $\hat{y}_t$ is the label predicted by the learner, the loss accrued in round $t$ is simply the squared error $\ell(\hat{y}_t, y_t) = (\hat{y}_t - y_t)^2$.

The notion of subsequence regret measured against a policy $\theta \in H$ in this setting leads exactly to the following groupwise regret metric for any given group $g$ (against $\theta$):

$$\text{Regret of Group } g \text{ against policy } \theta = \mathbb{E}\left[ \sum_{t=1}^T g(t)(\hat{y}_t - y_t)^2 \right] - \sum_{t=1}^T g(t)(\langle \theta, x_t \rangle - y_t)^2. \quad (1)$$

Our goal is to bound this quantity for each group $g$ in terms of $T_g := \sum_t g(t)$, which we refer to as the length of subsequence associated with group $g$, and in terms of the size of $\theta$. For each subsequence $I$, we use the online ridge regression algorithm of Azoury & Warmuth (2001) to instantiate the corresponding meta-expert $\mathcal{Z}_I$ in Algorithm 1. We note that our algorithm obtains the following groupwise regret guarantees as a direct corollary from Theorem 1.

**Corollary 1.** *There exists an efficient algorithm with groupwise regret guarantees for online linear regression with squared loss. This algorithm runs in time polynomial in $|G|$ and $d$ and guarantees that the regret for group $g$ as measured against any policy $\theta \in H$ is upper-bounded by*

$$O\left( d \ln(1 + T_g) + \sqrt{T_g \ln(|G|)} + \|\theta\|_2^2 \right),$$

*where $T_g$ denotes the length of group subsequence $g$.*

### 4.2 ONLINE CLASSIFICATION WITH GUARANTEES FOR MULTIPLE GROUPS

We now study online multi-label classification under multiple groups. We perform classification on an individual arriving in round $t$ based upon the individual's context $x_t$ and his group memberships. Formally, the context domain is $X \subseteq \{0, 1\}^d$ and each context describes an individual's feature vector; the prediction domain is $\mathcal{A} \subseteq \{0, 1\}^n$, with the prediction being a multi-dimensional label; and the adversary's action domain is $Y = \{0, 1\}^n$ with the action being the true multi-dimensional label that we aim to predict. Each subsequence $I$ corresponds to a group $g$, with the set of groups being denoted by $G = \{g_1, g_2, \cdots g_{|G|}\}$. We assume that the learner has access to an optimization oracle that can solve the corresponding, simpler, offline classification problem: i.e., given $t$ context-label pairs $\{x_s, y_s\}_{s=1}^t$, an oracle that finds the policy $f \in H$ that minimizes $\sum_{s=1}^t \ell(f(x_s), y_s)$ for the desired loss function $\ell(.)$.

Our subsequence regret is equivalently the groupwise regret associated with group $g$. Formally:

$$\text{Regret of Group } g = \mathbb{E}\left[ \sum_t g(t)\ell(\mathbf{p}_t(x_t), y_t) \right] - \min_{f \in H} \sum_t g(t)\ell\left(f(x_t), y_t\right).$$

---

[5]While these predictions may lie outside $[0, 1]$, we allow our instantiation of AdaNormalHedge to clip predictions outside this interval to the nearest endpoint to ensure the final prediction lies in $[0, 1]$, since this can only improve its performance. We also note that the algorithm of Azoury & Warmuth (2001) has reasonable bounds on the predictions, which is reflected in the corresponding external regret bounds (See Cesa-Bianchi & Lugosi (2006) for details).

We describe results for two special settings of this problem, introduced by Syrgkanis et al. (2016) for the external regret minimization version of this problem.

**Small Separator Sets:** A set $S \subset X$ of contexts is said to be a separator set of the policy class $H$ if for any two distinct policies $f, f' \in H$, there exists a context $x \in S$ such that $f(x) \neq f'(x)$. We present groupwise regret bounds assuming that $H$ has a separator set of size $s$ – note that a finite separator set implies that the set $H$ is also finite (upper bounded by $|\mathcal{A}|^s$).

**Transductive setting** In the transductive setting, the adversary reveals a set $S$ of contexts with $|S| = s$ to the learner before the sequence begins, along with the promise that all contexts in the sequence are present in the set $S$. Note that the order and multiplicity of the contexts given for prediction can be varied adaptively by the adversary.

When plugging in the online classification algorithm of Syrgkanis et al. (2016) as the regret minimizing algorithm $\mathcal{Z}_I$ for each subsequence $I$ (here, equivalently, for each group $g$) in Algorithm 1, we obtain the following efficiency and regret guarantees:

**Corollary 2.** *There exists an oracle efficient online classification algorithm (that runs in time polynomial in $|G|, d, n,$ and $s$) for classifiers with small separator sets (size $s$) with groupwise regret upper bounded by $O\left(\sqrt{T_g}(\sqrt{(s^{3/2}n^{3/2}\log|H|} + \sqrt{\log|G|})\right)$ for each group $g$.*

**Corollary 3.** *There exists an oracle efficient online classification algorithm (running in time polynomial in $|G|, d, n,$ and $s$) for classifiers in the transductive setting (with transductive set size $s$) with groupwise regret upper bounded by $O\left(\sqrt{T_g}(\sqrt{s^{1/2}n^{3/2}\log|H|} + \sqrt{\log|G|})\right)$ for each group $g$.*

Note that in the above corollaries, "oracle efficient" means an efficient reduction to a (batch) ERM problem—the above algorithms are efficient whenever the ERM problem can be solved efficiently. We also briefly mention how the ideas of Blum & Lykouris (2020) lift the results in this section for binary classification to the "apple-tasting model" where we only see the true label $y_t$ if the prediction is $p_t = 1$. For sake of brevity, this discussion is moved to Appendix G.

### 4.3 ONLINE LINEAR OPTIMIZATION

Our framework gives us subsequence regret guarantees for online linear optimization problems, which, among other applications, includes the online shortest path problem. In each round $t$, the algorithm picks a prediction $a_t$ from $\mathcal{A} \subseteq [0, 1]^d$ and the adaptive adversary picks cost vector $y_t$ from $Y \subseteq \mathbb{R}^d$ as its action. The algorithm obtains a loss of $\ell(a_t, y_t) = \langle a_t, y_t \rangle$. Unlike the previous applications, there is no context, and the policy class $H$ is directly defined as the set of all possible predictions $\mathcal{A}$. The algorithm is assumed to have access to an optimization oracle that finds some prediction in $\arg\min_{a \in \mathcal{A}} \langle a, c \rangle$ for any $c \in \mathbb{R}^d$. The objective is to bound the regret associated with each subsequence $I$ (described below) in terms of $T_I := \sum_t I(t)$, the length of the subsequence.

$$\text{Regret of Subsequence } I = \mathbb{E}\left[\sum_t I(t)\langle a_t, y_t \rangle\right] - \min_{a \in \mathcal{A}} \sum_t I(t)\langle a, y_t \rangle$$

Kalai & Vempala (2005) show an oracle efficient algorithm that has regret upper bounded by $\sqrt{8CAdT}$ where $A = \max_{a,a' \in \mathcal{A}} ||a - a'||_1$ and $C = \max_{c \in Y} ||c||_1$. Using this algorithm for minimizing external regret for each subsequence in Algorithm 1, we obtain the following corollary:

**Corollary 4.** *There is an oracle efficient algorithm for online linear optimization (of polynomial time in $|\mathcal{I}|$) with the regret of subsequence $I$ (of length $T_I$) upper bounded by $O(\sqrt{T_I CAd} + \sqrt{T_I \log|\mathcal{I}|})$.*

## 5 EXPERIMENTS

We empirically investigate the performance of our algorithm in an online regression problem. We present our experimental results on two datasets in the main body. First, we run experiments on a synthetic dataset in Section 5.2. Second, we perform experiments on real data in Section 5.3 using dataset (Lantz, 2013) for medical cost prediction tasks. We provide additional experiments on the census-based Adult income dataset [6] (Ding et al., 2021; Flood et al., 2020) in Appendix D.5.

---

[6]`adult_reconstruction.csv` is available at `https://github.com/socialfoundations/folktables`

## 5.1 ALGORITHM DESCRIPTION AND METRIC

**Learning task:** In our experiments, we run online linear regression with the goal of obtaining low subsequence regret. We focus on the case where subsequences are defined according to group membership, and we aim to obtain low regret in each group for the sake of fairness. Our loss functions and our regret metrics are the same as described in Section 4.1.

**Our algorithm:** We aim to evaluate the performance of Algorithm 1 for groupwise regret minimization. Given $|G|$ groups, our algorithm uses $|G| + 1$ subsequences: for each group $g \in G$, there is a corresponding subsequence that only includes the data of individuals in group $g$; further, we consider an additional "always active" subsequence that contains the entire data history. We recall that our algorithm uses an inner no-regret algorithm for each meta-expert or subsequence, and AdaNormal Hedge to choose across active subsequences on a given time step. We choose online ridge regression from Azoury & Warmuth (2001) for the meta-experts' algorithm as per Section 4.1.

**Baseline:** We compare our results to a simple baseline. At every time step $t$, the baseline uses the entire data history from the first $t - 1$ rounds, without splitting the data history across the $|G| + 1$ subsequences defined above. (Recall that since the subsequences intersect, there is no way to partition the data points by group). The baseline simply runs traditional online Ridge regression: at every $t$, it estimates $\hat{\theta}_t \triangleq \arg\min_\theta \{\sum_{\tau=1}^{t-1} (\langle \theta, x_\tau \rangle - y_\tau)^2 + \|\theta\|_2^2\}$, then predicts label $\hat{y}_t \triangleq \langle \hat{\theta}_t, x_t \rangle$.

Note that both our algorithm and the baseline have access to group identity (we concatenate group indicators to the original features), thus the baseline can learn models for different groups that differ by a linear shift[7].

## 5.2 SYNTHETIC DATA

**Feature and group generation** Our synthetic data is comprised of 5 groups: 3 shape groups (circle, square, triangle) and 2 color groups (red, green). Individual features are 20-dimensional; they are independently and identically distributed and taken from the uniform distribution over support $[0, 1]^{20}$. Each individual is assigned to one of the 3 shape groups and one of the 2 color groups randomly; this is described by vector $a_t \in \{0, 1\}^5$ which concatenates a categorical distribution over $p_{shape} = [0.5, 0.3, 0.2]$ and one over $p_{color} = [0.6, 0.4]$.

**Label generation** Individual labels are generated as follows: first, we draw a weight vector $w_g$ for each group $g$ uniformly at random on $[0, 1]^d$. Then, we assign an *intermediary label* to each individual in group $g$; this label is given by the linear expression $w_g^\top x$. Note that each individual has two intermediary labels by virtue of being in two groups: one from shape and one from color. The true label of an individual is then chosen to be a function of both intermediary labels, that we call the *label aggregation function*. We provide experiments from the following aggregation functions: mean of the intermediary labels; minimum of the intermediary labels; maximum of the intermediary labels; and another aggregation rule which we call the permutation model (described in Appendix D.3). Our main motivation for generating our labels in such a manner is the following: first, we pick linear models to make sure that linear regression has high performance in each group, which allows us to separate considerations of regret performance of our algorithm from considerations of performance of the model class we optimize on. Second, with high probability, the parameter vector $w_g$ significantly differs across groups; thus it is important for the algorithm to carefully ensemble models for individuals in the intersection of two groups, which is what makes it possible to out-perform the best linear models in hindsight and obtain negative regret.

**Results** Here, we show results for the mean aggregation function described in Section 5.2. Results for the other three (min, max, and permutation) are similar and provided in Appendix D.1,D.2,and D.3 respectively. Across all the groups we can see that our algorithm has significantly lower regret than the baseline, which in fact has linearly increasing regret. Note that this is not in conflict with the no-regret guarantee of Azoury & Warmuth (2001) as the best fixed linear model in hindsight is different for each group: indeed we see in box (f) that Azoury & Warmuth (2001) does have low regret on the entire sequence, despite having linearly increasing regret on every relevant subsequence. Further, we note that our algorithm's groupwise regret guarantees sometimes vastly outperform our already strong theoretical guarantees. In particular, while we show that our algorithm guarantees that the regret evolves sub-linearly, we note that in several of our experiments, it is in fact negative and decreasing over time (e.g. for the colors green and red). This means that our algorithm performs even better than the best linear regressor in hindsight; This is possible because by necessity, our algorithm

---

[7]We know of no implementable algorithms other than ours that has subsequence regret guarantees for linear regression (or other comparably large model class) that would provide an alternative point of comparison.

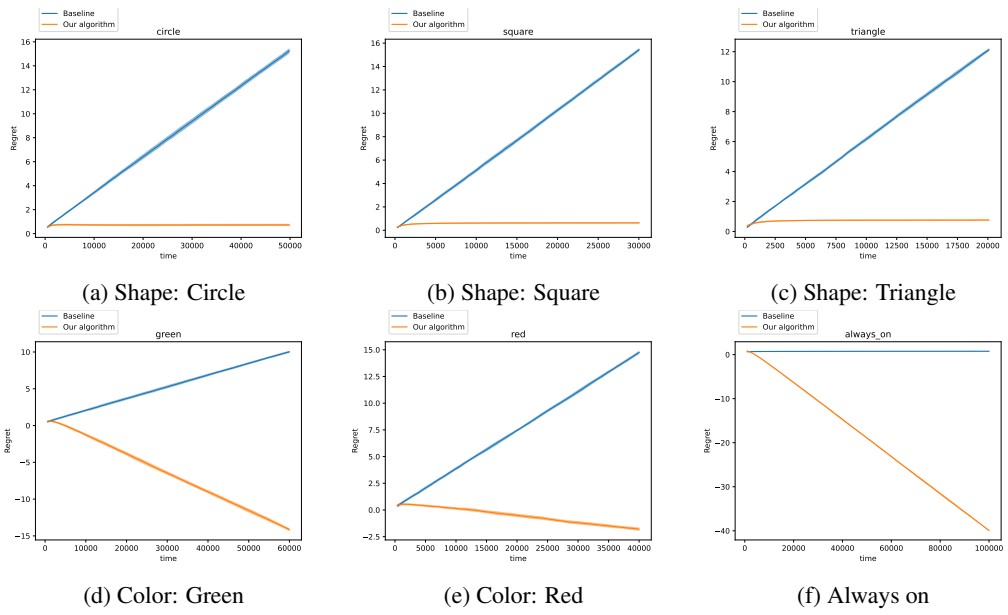

Figure 2: Regret for the baseline (blue) & our algorithm (orange) on synthetic data with mean of intermediary labels, our algorithm always has lower regret than the baseline.

is ensembling linear models for individuals who are members of multiple groups, giving it access to a more expressive (and more accurate) model class.

Quantitatively, the rough [8] magnitude of the cumulative least-squared loss (for the best performing linear model in hindsight) is 33. Our algorithm's cumulative loss is roughly lower than the baseline by an additive factor of 20, which represents a substantial improvement compared to the loss of the baseline. This implies that our regret improvements in Figure 2 are of a similar order of magnitude as the baseline loss itself, hence significant.

## 5.3 MEDICAL COST DATA

**Dataset description** The "Medical Cost" dataset Lantz (2013) looks at an individual medical cost prediction task. Individual features are split into 3 numeric {age, BMI, #children} and 3 categoric features {sex, smoker, region}. The dataset also has a real-valued medical charges column that we use as our label.

**Data pre-processing** All the numeric columns are min-max scaled to the range $[0, 1]$ and all the categoric columns are one-hot encoded. We also pre-process the data to limit the number of groups we are working with for simplicity of exposition. In particular, we simplify the following groups:

1. We divide age into 3 groups: young (age $\leq 35$), middle (age $\in (35, 50]$), old (age$> 50$)

2. For sex, we directly use the 2 groups given in the dataset : male, female

3. For smoker, we directly use the 2 groups given in the datset : smoker, non-smoker

4. We divide BMI (Body Masss Index) into 4 groups: underweight (BMI $< 18.5$), healthy (BMI $\in [18.5, 25)$), overweight (BMI $\in [25, 30)$), obese (BMI $\geq 30$)

**Results** The results for age, sex and smoker groups are discussed earlier, in Section 1, Figure 1. We remark that the BMI group results are similar: the groupwise regret is consistently lower and growing with time at a lower rate than the baseline. We provide detailed plots (including error bars) for all groups in Appendix D.4. Quantitatively, the rough magnitude[9] of the cumulative loss of the best linear model in hindsight is $\sim 4.1$, and our algorithm's cumulative loss is $\sim 1.9$ units lower than the baseline, which is a significant decrease.

---

[8]Exact values provided in Appendix E table 1

[9]Exact values provided in Appendix E table 5

## REPRODUCIBILITY STATEMENT

We briefly describe information about the data sequence, data processing and algorithms. The code is available at `https://github.com/krishnacharya/multidecomp`.
**Data sequence:** The sequence in which the online algorithms receive data can impact the empirical regret. We fix ten random seeds and use these to shuffle each of our datasets. The mean and standard deviation of regret across these shuffles is available in Appendix E.
**Dataset specific processing:** For the synthetic data we use a fixed seed (available in `README.md`) to generate the $100k$ samples, the aggregate labels are then min-max scaled to $[0, 1]$. For the two real datasets we min-max scale the numeric features to $[0, 1]$ and one-hot encode the categoric features.
**Algorithms:** Online ridge regression and AdaNormalHedge are implemented in `ORidge.py` and `Anh.py` respectively.

## ACKNOWLEDGMENTS

Supported in part by NSF grant IIS-2147212 and the Simons Collaboration on Algorithmic Fairness.

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

## A  ORGANIZATION

The Appendix is organized as follows. Appendix B contains a description of the sleeping experts setting and the AdaNormalHedge algorithm. Appendix C contains the proof of Theorem 1. Regret curves for the synthetic data with the 3 other label aggregations functions: min, max, permutation are in Appendix D.1, D.2 and D.3 respectively. In Appendix D.4 and D.5 we provide detailed plots for the 2 real datasets: Medical cost and Adult income respectively. Appendix E provides tables with regret at the end of each subsequence, along with cumulative loss of the best linear model for all the experiments. Appendix F compares including/excluding the always active expert in our algorithm. Lastly, Appendix G contains the binary classification with apple-tasting feedback results.

## B  SLEEPING EXPERTS, ADANORMALHEDGE

For completeness, here we describe the sleeping experts setting (Freund et al., 1997) and the AdaNormalHedge algorithm of Luo & Schapire (2015) with its regret guarantee. Infact, AdaNormalHedge provides a regret guarantee even in the confidence-rated setting i.e., $I(t) \in [0, 1]$ (Blum & Mansour, 2007; Luo & Schapire, 2015), a generalization of sleeping experts. The following is the interaction protocol in the sleeping experts setting.

1. At the start of each round $t$, each expert ($I \in \mathcal{I}$) reports if it's awake or asleep i.e., will it make a prediction ($I(t) = 1$) or not ($I(t) = 0$) [10].

2. The learner plays each expert according to a probability distribution $\mathbf{p}_t$, with the natural restriction that if the expert $I$ is asleep $p_{t,I} = 0$.

3. The loss $\ell_{t,I}$ for those experts who made predictions are revealed and the player suffers loss $\hat{\ell}_t = \langle \mathbf{p}_t, \boldsymbol{\ell}_t \rangle$. The instantaneous regret to expert $I$, $r_{t,I} \triangleq I(t)(\hat{\ell}_t - \ell_{t,I})$

---

[10]We overload $I$ to denote both the index of the sleeping expert as well as the indicator representing whether the expert is awake or not.

AdaNormalHedge (Sec 4, Luo & Schapire (2015)) sets the probability distribution

$$p_{t,I} \propto \frac{1}{|\mathcal{I}|} \cdot I(t) \cdot w(R_{t-1,I}, C_{t-1,I}) \tag{2}$$

Where $R_{t-1,I} = \sum_{\tau=1}^{t-1} r_{t,I}$ is the sum of instantaneous regret; and $C_{t-1,I} = \sum_{\tau=1}^{t-1} |r_{t,I}|$ is the sum of absolute values of instantaneous regret. The weight function $w(R,C) : \mathbb{R}^2 \to \mathbb{R}^+$ is defined as $w(R,C) \triangleq \frac{1}{2} \left( \exp\left( \frac{\max(0,R+1)^2}{3(C+1)} \right) - \exp\left( \frac{\max(0,R-1)^2}{3(C+1)} \right) \right)$. Note that at the start, i.e., $t = 0$, for all experts $R_{0,I} = C_{0,I} = 0$

With the probability set as in equation 2, AdaNormalHedge provides the following regret guarantee to each expert $I$, which scales sublinearly in the total number of times expert $I$ is awake:

$$\sum_{t=1}^{T} I(t) \cdot \langle \mathbf{p}_t, \boldsymbol{\ell}_t \rangle - \sum_{t=1}^{T} I(t) \cdot \ell_{t,I} \le O(\sqrt{T_I \log(|\mathcal{I}|)}) \tag{3}$$

## C  PROOF OF THEOREM 1

Our algorithm uses the AdaNormalHedge algorithm of Luo & Schapire (2015). AdaNormalHedge is an online prediction algorithm, that given a finite benchmark set of $|\mathcal{I}|$ policies and $|\mathcal{I}|$ time selection functions, guarantees a subsequence regret upper bound of $O(\sqrt{T_I \log |\mathcal{I}|})$ with a running time that is polynomial in $|\mathcal{I}|$ and the size of the contexts that arrive in each round. We refer to meta-experts $\{\boldsymbol{\mathcal{Z}}_I\}_{I \in \mathcal{I}}$ as policies even though they are themselves algorithms, and not policies mapping contexts to predictions, however, the output of each one of these algorithms in any time step are policies in $H$ mapping contexts to predictions. Therefore, when algorithm $\boldsymbol{\mathcal{Z}}$ picks $\boldsymbol{\mathcal{Z}}_I$ as the policy to play in a round, it is in fact picking the policy $z_t^I$ that is the output of algorithm $\boldsymbol{\mathcal{Z}}_I$ in that round.

Thus, the running time claim follows from the construction of Algorithm 1.

We introduce notation separating the independent random coins used by each of the algorithms used in our procedure. In the $t$-th round, each meta-expert algorithm $\boldsymbol{\mathcal{Z}}_I$ draws a randomized policy $\boldsymbol{p}_t^I$ using random coins $D^I$. The instantiation of ANH picks a policy $\boldsymbol{p}_t$ from among the above policies based upon random coins $D$. Each algorithm's random coins are independent of each other. By appealing to the regret guarantees of each algorithm $\boldsymbol{\mathcal{Z}}_I$ and taking the expectation only over the random coins used by these algorithms, we get the following for each subsequence $I$:

$$\left( \mathbb{E}_{D^I} \left[ \sum_t I(t) \ell(\boldsymbol{p}_t^I(x_t), y_t) \right] - \min_{f \in H} \sum_t I(t) \ell(f(x_t), y_t) \right) \le \alpha_I \tag{4}$$

where $\boldsymbol{p}_t^I$ is the (randomized) policy suggested by the algorithm $\boldsymbol{\mathcal{Z}}_I$ and $T_I = \sum_t I(t)$. Note that the actual prediction $\mathbf{p}_t(x_t)$ made by Algorithm 1 might differ from $\boldsymbol{p}_t^I(x_t)$, however since the regret guarantee holds for arbitrary adversarial sequences, we can still call upon it to measure the expected regret of hypothetically using this algorithm's prediction on the corresponding subsequence. Importantly, we update the algorithm $\boldsymbol{\mathcal{Z}}_I$ with the true label $y_t$ on every round only where the subsequence $I$ is active.

Next, we appeal to the regret guarantee of algorithm $\boldsymbol{\mathcal{Z}}$. Before doing so, we fix the realization of the offered expert $\boldsymbol{p}_t^I$ as $z_t^I$ from each meta-expert $\boldsymbol{\mathcal{Z}}_I$ i.e. we do the analysis conditioned upon the draw of internal randomness used by each meta-expert. In particular, our analysis holds for every possible realization of predictions $\boldsymbol{p}_t^I$ offered by each of the meta-experts over all the rounds. Since our algorithm updates the loss of each meta-expert with respect to these realized draws of the meta-experts, therefore, we use the subsequence regret guarantee of $\boldsymbol{\mathcal{Z}}$ (i.e the results about AdaNormalHedge in Luo & Schapire (2015)) for each subsequence $I$ to get:

$$\left( \mathbb{E}_D \left[ \sum_t I(t) \ell(\mathbf{p}_t(x_t), y_t) \right] - \sum_t I(t) \ell(z_t^I(x_t), y_t) \right) \le O\left( \sqrt{T_I \log(|\mathcal{I}|)} \right) \tag{5}$$

where the expectation is taken *only* over the random coins $D$ used by $\mathcal{A}$ to select which meta-expert to use in each round. Since the inequality 5 holds for every realization of the randomness used by the meta-experts, the same inequality holds in expectation over the random coins $D^I$ used by the corresponding meta-expert:

$$\left( \mathbb{E}_{(D \otimes D^I)} \left[ \sum_t I(t) \ell(\mathbf{p}_t(x_t), y_t) \right] - \mathbb{E}_{D^I} \left[ \sum_t I(t) \ell(\mathbf{p}_t^I(x_t), y_t) \right] \right) \le O\left( \sqrt{T_I \log(|\mathcal{I}|)} \right) \quad (6)$$

Inequality 4 will also hold when we additionally take the expectation over the random coins $D$ used by the copy of ANH, since $D^I$ is independent of $D$.

Thus, putting together 4and 6 in this proof gives us (for each subsequence $I$):

$$\left( \mathbb{E}_{(\otimes_I D^I) \otimes D} \left[ \sum_t I(t) \ell(\mathbf{p}_t(x_t), y_t) \right] - \min_{f \in H} \sum_t I(t) \ell(f(x_t), y_t) \right) \le \alpha_I + O\left( \sqrt{T_I \log(|\mathcal{I}|)} \right)$$

where the expectation is taken over the product distributions of random coins used by all the algorithms.

## D  ADDITIONAL FIGURES

On the synthetic data with the 3 other aggregation rules: min, max, permutation; the plots and qualitative inferences are similar to the mean aggregation in Section 5.2, i.e., across all the groups we see that our algorithm has significantly lower regret than the baseline, which in fact has linearly increasing regret [11]. Thus, here, we only discuss quantitative values such as the magnitudes of the cumulative loss, and difference between the cumulative loss of the baseline and our algorithm.

### D.1  SYNTHETIC DATA - MIN

Recall here that the label of any individual is the minimum of the intermediary labels of the groups it's a part of. Quantitatively, the rough magnitude [12] of the cumulative loss of the best linear model in hindsight is $\sim 79$, and our algorithm's cumulative loss is $\sim 48$ units lower than the baseline, which is a significant decrease.

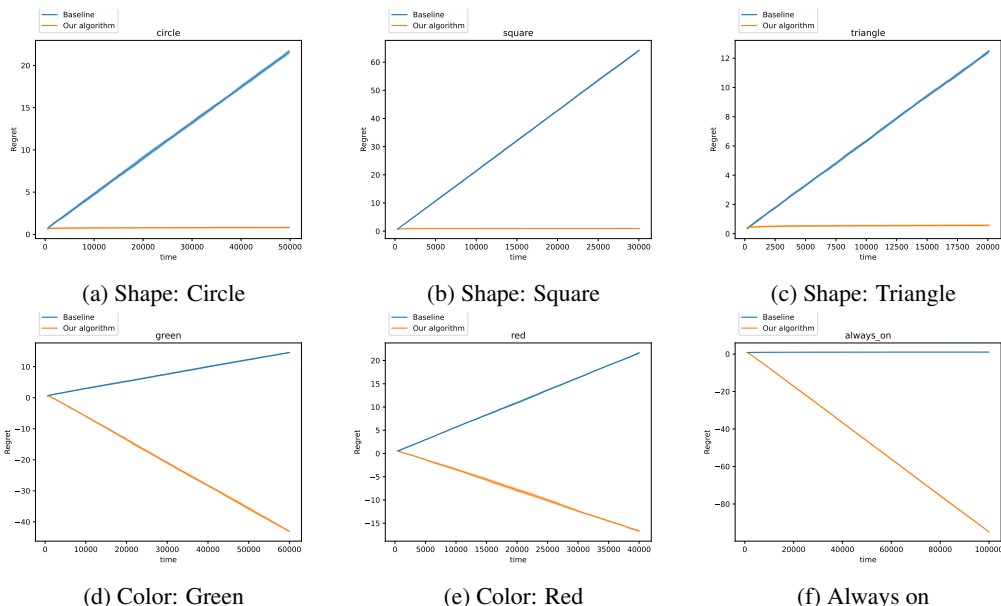

(a) Shape: Circle    (b) Shape: Square    (c) Shape: Triangle

(d) Color: Green    (e) Color: Red    (f) Always on

Figure 3: Regret for the baseline (blue) & our algorithm (orange) on synthetic data with minimum of intermediary labels, our algorithm always has lower regret than the baseline.

### D.2  SYNTHETIC DATASET - MAX

Recall here that the label of any individual is the maximum of the intermediary labels of the groups it's a part of. Quantitatively, the rough magnitude [13] of the cumulative loss of the best linear model in

---

[11]On the always on group, the baseline has sublinear growth, we have discussed this phenomenon in Sec 5.2

[12]Exact values provided in Appendix E table 2

[13]Exact values provided in Appendix E table 3

hindsight is $\sim 59$, and our algorithm's cumulative loss is $\sim 34$ units lower than the baseline, which is a significant decrease.

(a) Shape: Circle  (b) Shape: Square  (c) Shape: Triangle

(d) Color: Green  (e) Color: Red  (f) Always on

Figure 4: Regret for the baseline (blue) & our algorithm (orange) on synthetic data with maximum of intermediary labels, our algorithm always has lower regret than the baseline.

### D.3 SYNTHETIC DATASET - PERMUTATION

Recall here there is an underlying fixed ordering or permutation over the groups, and the label of any individual is decided by the linear model corresponding to the *first* group in the ordering that this individual is a part of. The permutation (chosen randomly) for experiments in Figure 5 is (1:green, 2:square, 3:red, 4:triangle, 5:circle). Let's see two simple examples to understand this aggregation: For an individual which is a red square we use the intermediary label for square, for an individual which is a green square we use the intermediary label for green.

Quantitatively, the rough magnitude [14] of the cumulative loss of the best linear model in hindsight is $\sim 94$, and our algorithm's cumulative loss is $\sim 93$ units lower than the baseline, which is a significant decrease.

---

[14]Exact values provided in Appendix E table 4

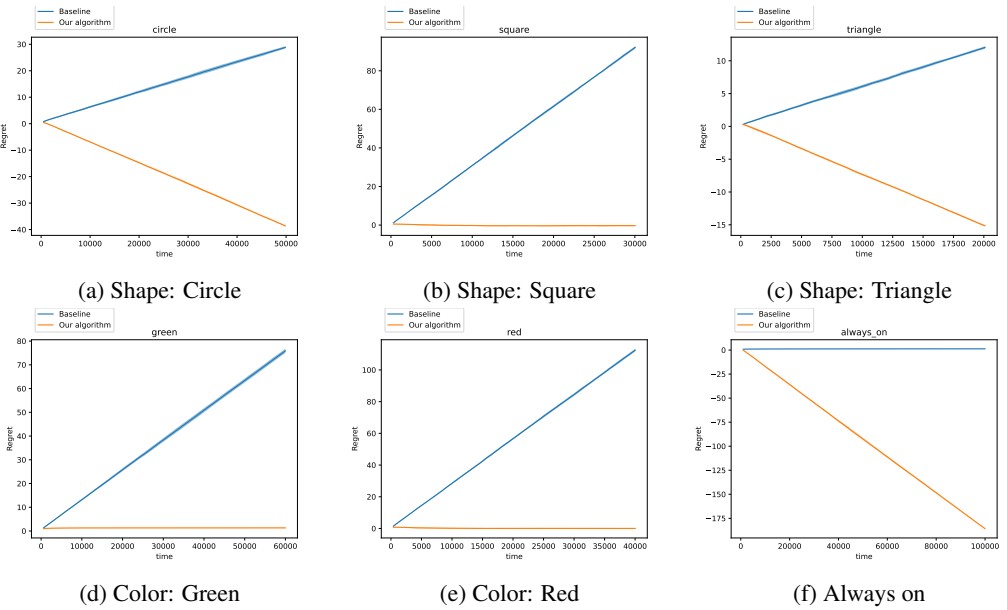

(a) Shape: Circle      (b) Shape: Square      (c) Shape: Triangle

(d) Color: Green      (e) Color: Red      (f) Always on

Figure 5: Regret for the baseline (blue) & our algorithm (orange) on synthetic data with permutation aggregation of intermediary labels, our algorithm always has lower regret than the baseline

## D.4 MEDICAL COSTS DATASET

For the medical cost data we have already discussed the rough quantitative results in Section 5.3, for exact values see Appendix E table 5. Here we discuss regret curves for the baseline and our algorithm on all the groups.

**Regret on age, sex groups** On all three age groups: young, middle, old (Fig 6), and both sex groups: male, female (Fig 7); our algorithm has significantly lower regret than the baseline, in fact, our algorithm's regret is negative and decreasing over time. i.e., our algorithm outperforms the best linear regressor in hindsight. This qualitatively matches the same phenomenon occurring in the synthetic data.

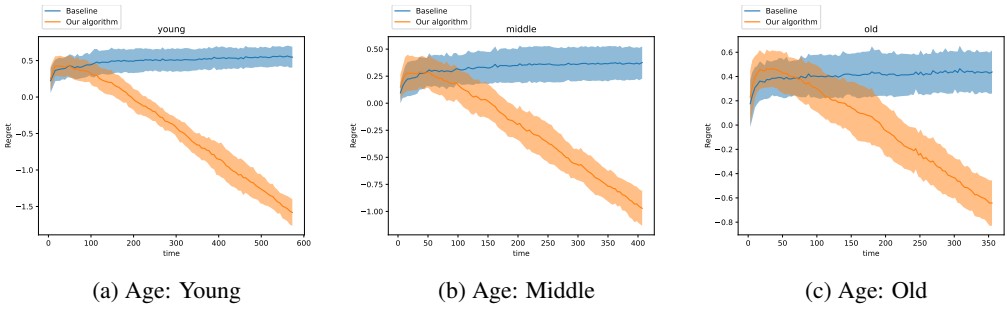

(a) Age: Young      (b) Age: Middle      (c) Age: Old

Figure 6: Age groups

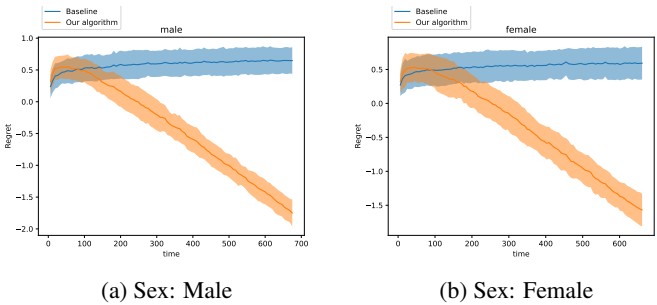

(a) Sex: Male      (b) Sex: Female

Figure 7: Sex groups

**Regret on smoker, bmi groups**    For both smokers and non smokers (Fig 8), and all four bmi groups: underweight, healthyweight, overweight, obese (Fig 9); our algorithm has significantly lower regret than the baseline, which in fact has linearly increasing regret.

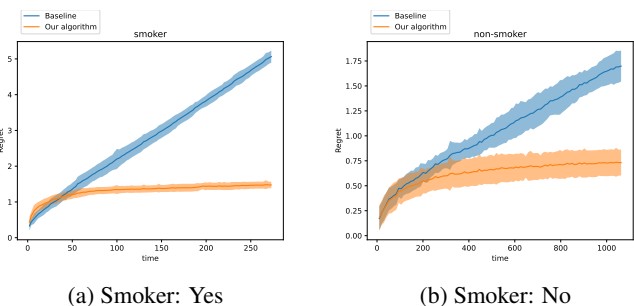

(a) Smoker: Yes      (b) Smoker: No

Figure 8: Smoking groups

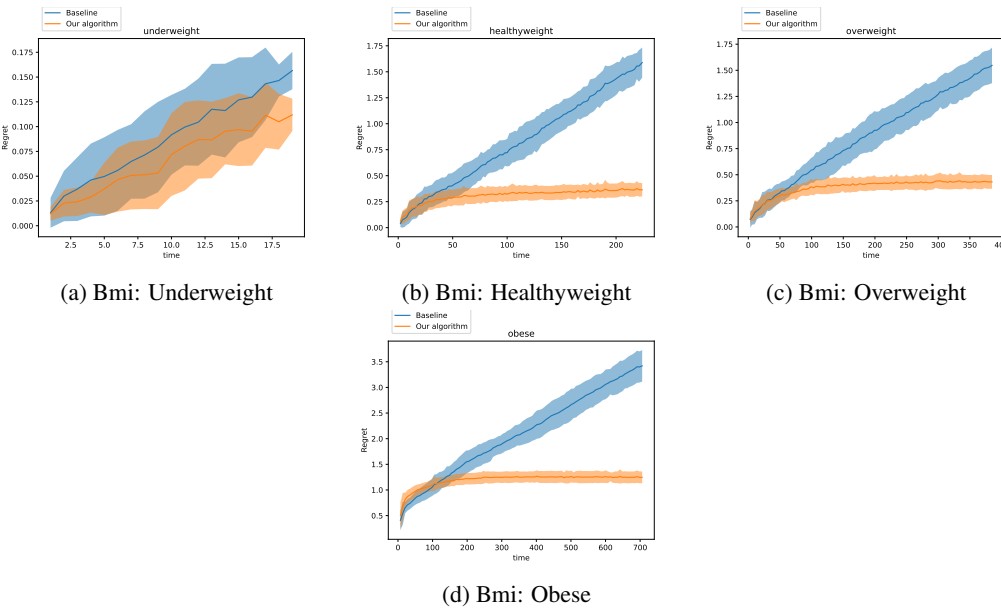

(a) Bmi: Underweight      (b) Bmi: Healthyweight      (c) Bmi: Overweight

(d) Bmi: Obese

Figure 9: Bmi groups

**Regret on the always-on group**    In Fig 10 our algorithm's regret is negative and decreasing with time, i.e., it outperforms the best linear regressor in hindsight. The baseline's regret evolves sublinearly, this matches the same phenomenon occurring in the synthetic data in Section 5.2 Fig 2 (f).

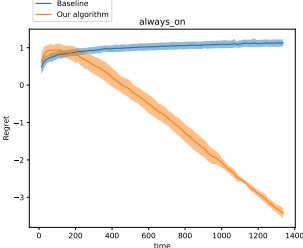

Figure 10: Always on

## D.5 ADULT INCOME DATASET

**Dataset description** The Adult income dataset [15] is targeted towards income prediction based on census data. Individual features are split into: 5 numeric features, {`hours-per-week`, `age`, `capital-gain`, `capital-loss`, `education-num`}; and 8 categoric features, {`workclass`, `education`, `marital-status`, `relationship`, `race`, `sex`, `native-country`, `occupation`}. The dataset also contains a real-valued `income` column that we use as our label.

**Data pre-processing** All the numeric columns are min-max scaled to $[0, 1]$, while all the categoric columns are one-hot encoded

We also pre-process the data to limit the number of groups we are working with for simplicity of exposition. In particular, we simplify the following groups:

1. We divide the age category into three groups: young (age $\leq 35$), middle (age $\in (35, 50]$), old (age$> 50$)

2. We divide the education level into 2 groups: at most high school versus college or more.

3. For sex, we directly use the 2 groups given in the dataset: male, female.

4. For race, we directly use the 5 groups given in the dataset: White, Black, Asian-Pac-Islander, Amer-Indian-Eskimo, Other.

Quantitatively, the rough magnitude [16] of cumulative loss of the best linear model in hindsight is $\sim 568$, and our algorithm's cumulative loss is $\sim 14$ units lower than the baseline, so there is a decrease but not as significant as for the other datasets. This may be because in this case, conditional on having similar features, group membership is not significantly correlated with income; in fact, we observe in our data that different groups have relatively similar best fitting models. This explains why a one-size-fits-all external regret approach like Azoury & Warmuth (2001) has good performance: one can use the same model for all groups and does not need to fit a specific, different model to each group.

**Regret on age groups** On all three age groups: young, middle, old (Fig 11); our algorithm has lower regret than the baseline.

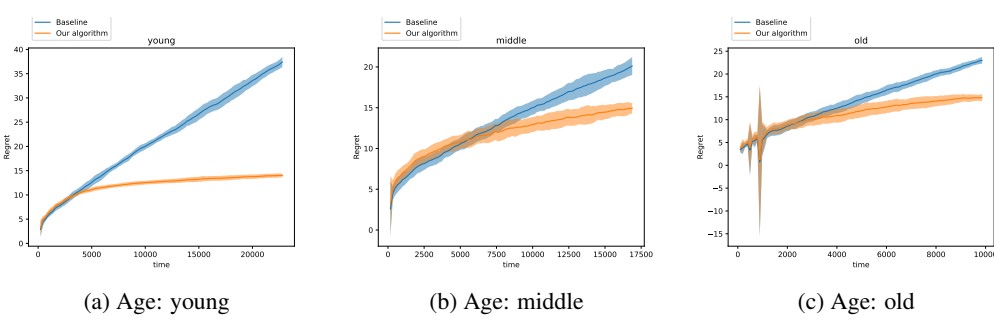

(a) Age: young          (b) Age: middle          (c) Age: old

Figure 11: Age groups

**Regret on education level groups** For both the education level groups: atmost high school and college or more (Fig 12), our algorithm has lower regret than the baseline.

---

[15] adult_reconstruction.csv at `https://github.com/socialfoundations/folktables`
[16] Exact values provided in Appendix E table 6

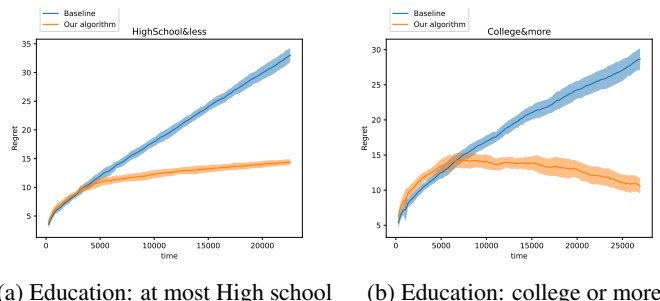

(a) Education: at most High school    (b) Education: college or more

Figure 12: Education level groups

**Regret on sex groups** On both the sex groups: male and female (Fig 13), our algorithm has lower regret than the baseline

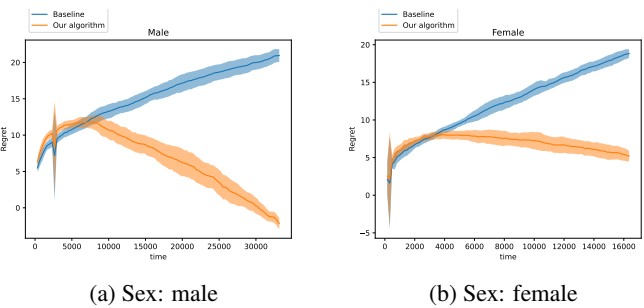

(a) Sex: male    (b) Sex: female

Figure 13: Sex groups

**Regret on race groups** On all 5 race groups: White, Black, Asian-Pac-Islander, Amer-Indian-Eskimo, Other (Fig 14) our algorithm has lower regret than the baseline.

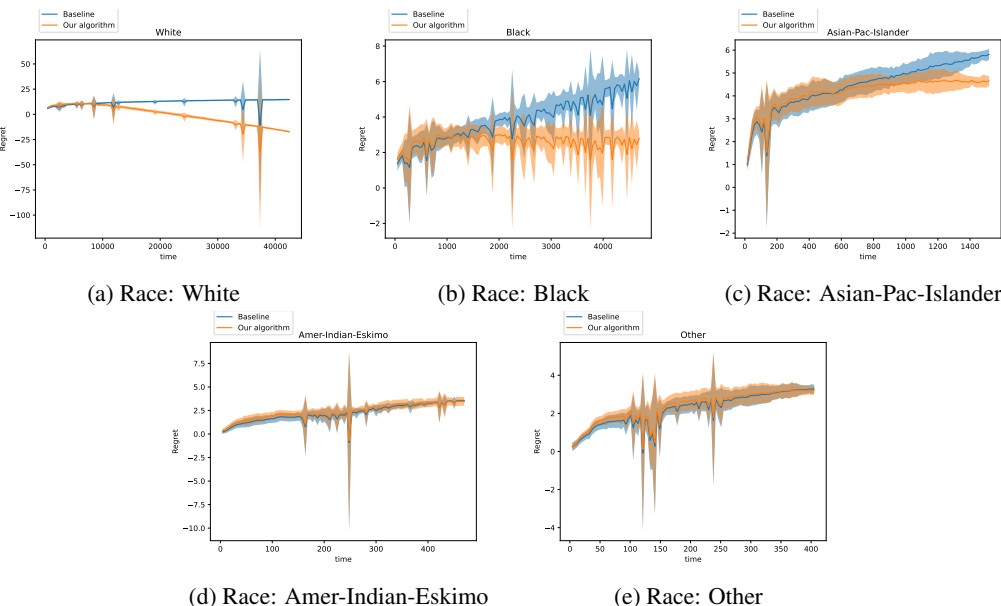

(a) Race: White    (b) Race: Black    (c) Race: Asian-Pac-Islander

(d) Race: Amer-Indian-Eskimo    (e) Race: Other

Figure 14: Race groups

**Regret on always-on group** In Fig 15 our algorithm's regret is negative and decreasing with time, i.e., it outperforms the best linear regressor in hindsight. The baseline's regret evolves sublinearly, this matches the same phenomenon occurring in the synthetic data in Section 5.2 Fig 2 (f).

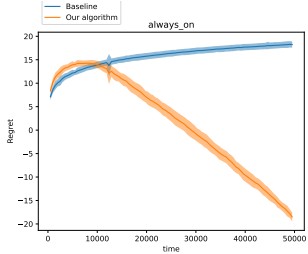

Figure 15: Always on

## E TABLES

In these tables, for a given group we record: its size, regret for the baseline, regret for our algorithm, and the cumulative loss of the best linear model in hindsight. Since the baseline and our algorithm are online algorithms we feed the data one-by-one using 10 random[17] shuffles, recording the mean and standard deviation. Note that the cumulative loss of the best linear model in hindsight will be the same across all the shuffles, since it's just solving for the least squares solution on the entire group subsequence.

| Group name | Group size | Baseline's regret | Our algorithm's regret | Cumulative loss of best linear model in hindsight |
|---|---|---|---|---|
| circle | 49857 | $15.25 \pm 0.22$ | $0.73 \pm 0.09$ | 23.57 |
| square | 30044 | $15.43 \pm 0.13$ | $0.63 \pm 0.06$ | 14.12 |
| triangle | 20099 | $12.13 \pm 0.10$ | $0.76 \pm 0.04$ | 9.43 |
| green | 59985 | $10.02 \pm 0.15$ | $-14.13 \pm 0.19$ | 38.81 |
| red | 40015 | $14.75 \pm 0.16$ | $-1.80 \pm 0.17$ | 26.37 |
| always_on | 100000 | $0.76 \pm 0.06$ | $-39.93 \pm 0.10$ | 89.18 |

Table 1: Synthetic data with mean of intermediary labels

| Group name | Group size | Baseline's regret | Our algorithm's regret | Cumulative loss of best linear model in hindsight |
|---|---|---|---|---|
| circle | 49857 | $21.62 \pm 0.28$ | $0.81 \pm 0.10$ | 63.49 |
| square | 30044 | $64.21 \pm 0.31$ | $0.94 \pm 0.05$ | 15.17 |
| triangle | 20099 | $12.47 \pm 0.13$ | $0.57 \pm 0.07$ | 26.37 |
| green | 59985 | $14.55 \pm 0.19$ | $-43.10 \pm 0.23$ | 97.42 |
| red | 40015 | $21.64 \pm 0.20$ | $-16.69 \pm 0.23$ | 69.72 |
| always_on | 100000 | $1.04 \pm 0.06$ | $-94.94 \pm 0.08$ | 202.29 |

Table 2: Synthetic data with min of intermediary labels

---

[17]available at `https://github.com/krishnacharya/multidecomp`

| Group name | Group size | Baseline's regret | Our algorithm's regret | Cumulative loss of best linear model in hindsight |
|---|---|---|---|---|
| circle | 49857 | $22.78 \pm 0.32$ | $0.91 \pm 0.07$ | 32.46 |
| square | 30044 | $17.86 \pm 0.14$ | $0.52 \pm 0.08$ | 36.92 |
| triangle | 20099 | $29.51 \pm 0.22$ | $0.87 \pm 0.05$ | 9.42 |
| green | 59985 | $8.65 \pm 0.20$ | $-23.25 \pm 0.24$ | 65.38 |
| red | 40015 | $12.48 \pm 0.22$ | $-23.46 \pm 0.29$ | 62.42 |
| always_on | 100000 | $0.86 \pm 0.08$ | $-66.98 \pm 0.13$ | 148.08 |

Table 3: Synthetic data with max of intermediary labels

| Group name | Group size | Baseline's regret | Our algorithm's regret | Cumulative loss of best linear model in hindsight |
|---|---|---|---|---|
| circle | 49857 | $28.84 \pm 0.45$ | $-38.63 \pm 0.17$ | 77.14 |
| square | 30044 | $92.10 \pm 0.64$ | $-0.28 \pm 0.27$ | 54.87 |
| triangle | 20099 | $12.03 \pm 0.20$ | $-15.15 \pm 0.09$ | 30.67 |
| green | 59985 | $75.89 \pm 0.89$ | $1.31 \pm 0.07$ | 28.95 |
| red | 40015 | $112.49 \pm 0.88$ | $0.04 \pm 0.26$ | 78.32 |
| always_on | 100000 | $1.26 \pm 0.09$ | $-185.77 \pm 0.25$ | 294.39 |

Table 4: Synthetic data with permutation of intermediary labels

| Group name | Group size | Baseline's regret | Our algorithm's regret | Cumulative loss of best linear model in hindsight |
|---|---|---|---|---|
| young | 574 | $0.55 \pm 0.14$ | $-1.58 \pm 0.18$ | 5.40 |
| middle | 408 | $0.38 \pm 0.15$ | $-0.97 \pm 0.16$ | 3.34 |
| old | 356 | $0.44 \pm 0.18$ | $-0.64 \pm 0.19$ | 3.13 |
| underweight | 20 | $0.16 \pm 0.02$ | $0.11 \pm 0.02$ | 0.03 |
| healthyweight | 225 | $1.59 \pm 0.14$ | $0.36 \pm 0.07$ | 1.00 |
| overweight | 386 | $1.55 \pm 0.17$ | $0.43 \pm 0.07$ | 1.84 |
| obese | 707 | $3.42 \pm 0.31$ | $1.25 \pm 0.12$ | 3.65 |
| smoker | 274 | $5.06 \pm 0.17$ | $1.47 \pm 0.09$ | 0.90 |
| non-smoker | 1064 | $1.70 \pm 0.15$ | $0.73 \pm 0.13$ | 5.57 |
| male | 676 | $0.65 \pm 0.21$ | $-1.75 \pm 0.21$ | 6.11 |
| female | 662 | $0.59 \pm 0.24$ | $-1.57 \pm 0.24$ | 5.88 |
| always_on | 1338 | $1.13 \pm 0.09$ | $-3.43 \pm 0.13$ | 12.10 |

Table 5: Medical costs data

| Group name | Group size | Baseline's regret | Our algorithm's regret | Cumulative loss of best linear model in hindsight |
|---|---|---|---|---|
| young | 22792 | $37.40 \pm 1.04$ | $14.01 \pm 0.41$ | 421.89 |
| middle | 16881 | $20.11 \pm 1.11$ | $14.93 \pm 0.63$ | 608.15 |
| old | 9858 | $22.99 \pm 0.74$ | $14.81 \pm 0.75$ | 414.61 |
| HighSchool&less | 22584 | $33.02 \pm 1.21$ | $14.38 \pm 0.51$ | 499.55 |
| College&more | 26947 | $28.67 \pm 1.54$ | $10.55 \pm 1.08$ | 963.92 |
| Male | 33174 | $20.95 \pm 0.86$ | $-2.17 \pm 0.77$ | 1171.06 |
| Female | 16357 | $18.82 \pm 0.54$ | $5.19 \pm 0.69$ | 314.32 |
| White | 42441 | $14.67 \pm 0.76$ | $-17.42 \pm 0.98$ | 1325.50 |
| Asian-Pac-Islander | 1519 | $5.82 \pm 0.23$ | $4.66 \pm 0.21$ | 55.05 |
| Amer-Indian-Eskimo | 471 | $3.55 \pm 0.12$ | $3.48 \pm 0.44$ | 9.04 |
| Other | 406 | $3.28 \pm 0.24$ | $3.21 \pm 0.24$ | 7.50 |
| Black | 4694 | $6.16 \pm 0.39$ | $2.79 \pm 0.36$ | 94.58 |
| always_on | 49531 | $18.25 \pm 0.66$ | $-18.51 \pm 0.94$ | 1506.92 |

Table 6: Adult income data

## F EFFECT OF ADDING AN ALWAYS ACTIVE EXPERT OR NOT ON OUR ALGORITHM

Throughout in the earlier sections, we include an "always on" expert in Algorithm 1, as in terms of theory, doing so provides a regret bound over the entire sequence, which does not follow generically from having regret bounds for subsequences[18]. Here in tables 7 - 12 we compare the regret of our algorithm when including/excluding the "always on" expert and see that the effect is marginal, with both these variants being much better than the baseline.

| Group name | Group size | Baseline's regret | Regret of our algorithm including always active | Regret of our algorithm excluding always active | Cuml loss of best linear model |
|---|---|---|---|---|---|
| circle | 49857 | $15.25 \pm 0.22$ | $0.73 \pm 0.09$ | $0.66 \pm 0.09$ | 23.57 |
| square | 30044 | $15.43 \pm 0.13$ | $0.63 \pm 0.06$ | $0.56 \pm 0.06$ | 14.12 |
| triangle | 20099 | $12.13 \pm 0.10$ | $0.76 \pm 0.04$ | $0.73 \pm 0.04$ | 9.43 |
| green | 59985 | $10.02 \pm 0.15$ | $-14.13 \pm 0.19$ | $-14.23 \pm 0.16$ | 38.81 |
| red | 40015 | $14.75 \pm 0.16$ | $-1.80 \pm 0.17$ | $-1.86 \pm 0.14$ | 26.37 |
| always_on | 100000 | $0.76 \pm 0.06$ | $-39.93 \pm 0.10$ | $-40.10 \pm 0.09$ | 89.18 |

Table 7: Synthetic data with mean of intermediary labels

| Group name | Group size | Baseline's regret | Regret of our algorithm including always active | Regret of our algorithm excluding always active | Cuml loss of best linear model |
|---|---|---|---|---|---|
| circle | 49857 | $21.62 \pm 0.28$ | $0.81 \pm 0.10$ | $0.77 \pm 0.10$ | 63.49 |
| square | 30044 | $64.21 \pm 0.31$ | $0.94 \pm 0.05$ | $0.83 \pm 0.05$ | 15.17 |
| triangle | 20099 | $12.47 \pm 0.13$ | $0.57 \pm 0.07$ | $0.60 \pm 0.06$ | 26.37 |
| green | 59985 | $14.55 \pm 0.19$ | $-43.10 \pm 0.23$ | $-43.19 \pm 0.21$ | 97.42 |
| red | 40015 | $21.64 \pm 0.20$ | $-16.69 \pm 0.23$ | $-16.72 \pm 0.21$ | 69.72 |
| always_on | 100000 | $1.04 \pm 0.06$ | $-94.94 \pm 0.08$ | $-95.06 \pm 0.08$ | 202.29 |

Table 8: Synthetic data with min of intermediary labels

---

[18]If there are subsequences that partition the entire sequence, then one gets a regret bound on the entire sequence that follows from summing up the regret bounds on the partition, but this is wasteful. If the subsequences do not partition the space, then there is not necessarily a regret bound on the entire sequence without including the "always on" expert

| Group name | Group size | Baseline's regret | Regret of our algorithm including always active | Regret of our algorithm excluding always active | Cuml loss of best linear model |
|---|---|---|---|---|---|
| circle | 49857 | $22.78 \pm 0.32$ | $0.91 \pm 0.07$ | $0.86 \pm 0.07$ | 32.46 |
| square | 30044 | $17.86 \pm 0.14$ | $0.52 \pm 0.08$ | $0.45 \pm 0.07$ | 36.92 |
| triangle | 20099 | $29.51 \pm 0.22$ | $0.87 \pm 0.05$ | $0.80 \pm 0.05$ | 9.42 |
| green | 59985 | $8.65 \pm 0.20$ | $-23.25 \pm 0.24$ | $-23.30 \pm 0.23$ | 65.38 |
| red | 40015 | $12.48 \pm 0.22$ | $-23.46 \pm 0.29$ | $-23.59 \pm 0.28$ | 62.42 |
| always_on | 100000 | $0.86 \pm 0.08$ | $-66.98 \pm 0.13$ | $-67.17 \pm 0.12$ | 148.08 |

Table 9: Synthetic data with max of intermediary labels

| Group name | Group size | Baseline's regret | Regret of our algorithm including always active | Regret of our algorithm excluding always active | Cuml loss of best linear model |
|---|---|---|---|---|---|
| circle | 49857 | $28.84 \pm 0.45$ | $-38.63 \pm 0.17$ | $-38.66 \pm 0.21$ | 77.14 |
| square | 30044 | $92.10 \pm 0.64$ | $-0.28 \pm 0.27$ | $-0.33 \pm 0.32$ | 54.87 |
| triangle | 20099 | $12.03 \pm 0.20$ | $-15.15 \pm 0.09$ | $-15.11 \pm 0.08$ | 30.67 |
| green | 59985 | $75.89 \pm 0.89$ | $1.31 \pm 0.07$ | $1.27 \pm 0.07$ | 28.95 |
| red | 40015 | $112.49 \pm 0.88$ | $0.04 \pm 0.26$ | $0.05 \pm 0.27$ | 78.32 |
| always_on | 100000 | $1.26 \pm 0.09$ | $-185.77 \pm 0.25$ | $-185.81 \pm 0.25$ | 294.39 |

Table 10: Synthetic data with permutation of intermediary labels

| Group name | Group size | Baseline's regret | Regret of our algorithm including always active | Regret of our algorithm excluding always active | Cuml loss of best linear model |
|---|---|---|---|---|---|
| young | 574 | $0.55 \pm 0.14$ | $-1.58 \pm 0.18$ | $-1.63 \pm 0.18$ | 5.40 |
| middle | 408 | $0.38 \pm 0.15$ | $-0.97 \pm 0.16$ | $-0.99 \pm 0.17$ | 3.34 |
| old | 356 | $0.44 \pm 0.18$ | $-0.64 \pm 0.19$ | $-0.64 \pm 0.19$ | 3.13 |
| underweight | 20 | $0.16 \pm 0.02$ | $0.11 \pm 0.02$ | $0.11 \pm 0.02$ | 0.03 |
| healthyweight | 225 | $1.59 \pm 0.14$ | $0.36 \pm 0.07$ | $0.34 \pm 0.06$ | 1.00 |
| overweight | 386 | $1.55 \pm 0.17$ | $0.43 \pm 0.07$ | $0.40 \pm 0.06$ | 1.84 |
| obese | 707 | $3.42 \pm 0.31$ | $1.25 \pm 0.12$ | $1.24 \pm 0.12$ | 3.65 |
| smoker | 274 | $5.06 \pm 0.17$ | $1.47 \pm 0.09$ | $1.43 \pm 0.09$ | 0.90 |
| non-smoker | 1064 | $1.70 \pm 0.15$ | $0.73 \pm 0.13$ | $0.71 \pm 0.14$ | 5.57 |
| male | 676 | $0.65 \pm 0.21$ | $-1.75 \pm 0.21$ | $-1.79 \pm 0.21$ | 6.11 |
| female | 662 | $0.59 \pm 0.24$ | $-1.57 \pm 0.24$ | $-1.59 \pm 0.24$ | 5.88 |
| always_on | 1338 | $1.13 \pm 0.09$ | $-3.43 \pm 0.13$ | $-3.49 \pm 0.13$ | 12.10 |

Table 11: Medical costs data

| Group name | Group size | Baseline's regret | Regret of our algorithm including always active | Regret of our algorithm excluding always active | Cuml loss of best linear model |
|---|---|---|---|---|---|
| young | 22792 | $37.40 \pm 1.04$ | $14.01 \pm 0.41$ | $13.79 \pm 0.40$ | 421.89 |
| middle | 16881 | $20.11 \pm 1.11$ | $14.93 \pm 0.63$ | $14.70 \pm 0.82$ | 608.15 |
| old | 9858 | $22.99 \pm 0.74$ | $14.81 \pm 0.75$ | $14.59 \pm 0.97$ | 414.61 |
| HighSchool&less | 22584 | $33.02 \pm 1.21$ | $14.38 \pm 0.51$ | $14.05 \pm 0.42$ | 499.55 |
| College&more | 26947 | $28.67 \pm 1.54$ | $10.55 \pm 1.08$ | $10.21 \pm 1.03$ | 963.92 |
| Male | 33174 | $20.95 \pm 0.86$ | $-2.17 \pm 0.77$ | $-3.46 \pm 0.63$ | 1171.06 |
| Female | 16357 | $18.82 \pm 0.54$ | $5.19 \pm 0.69$ | $5.81 \pm 0.81$ | 314.32 |
| White | 42441 | $14.67 \pm 0.76$ | $-17.42 \pm 0.98$ | $-19.10 \pm 0.93$ | 1325.50 |
| Asian-Pac-Islander | 1519 | $5.82 \pm 0.23$ | $4.66 \pm 0.21$ | $5.06 \pm 0.42$ | 55.05 |
| Amer-Indian-Eskimo | 471 | $3.55 \pm 0.12$ | $3.48 \pm 0.44$ | $3.80 \pm 0.41$ | 9.04 |
| Other | 406 | $3.28 \pm 0.24$ | $3.21 \pm 0.24$ | $3.41 \pm 0.27$ | 7.50 |
| Black | 4694 | $6.16 \pm 0.39$ | $2.79 \pm 0.36$ | $2.88 \pm 0.44$ | 94.58 |
| always_on | 49531 | $18.25 \pm 0.66$ | $-18.51 \pm 0.94$ | $-19.18 \pm 0.99$ | 1506.92 |

Table 12: Adult income data

## G   APPLE TASTING MODEL

In the apple tasting model for binary classification ($n = 1$), we only see the label $y_t$ at the end of the $t$-th round if the prediction $p_t$ is to "accept", i.e.,$p_t = 1$. We present results in a generalization of this model, introduced by Blum & Lykouris (2020), called the Pay-for-feedback model. In this model, the algorithm is allowed to pay the maximum possible loss of $1$ at the end of the $t$-th round to see the label $y_t$. The sum of all the payments is added to the regret expression to create a new objective that we wish to upper bound. For some notation - $v_t$ is the indicator variable that is set to 1 if we pay to view the label at the end of round $t$. This new objective is written down below -

$$\text{Groupwise Regret with Pay-for-feeback} = \left( \mathbb{E}\left[ \sum_t g_i(t)(\ell(\boldsymbol{p}_t, y_t) + \boldsymbol{v}_t) \right] - \min_{f \in F_i} \sum_t g_i(t)\ell(f(x_t), y_t) \right)$$

Blum & Lykouris (2020) introduce a blackbox method that converts any online algorithm with regret for group $g$ upper bounded by $O(\sqrt{T_g})$ (other terms not dependent on $T_g$ can multiply with this expression) into an algorithm with a corresponding bound in the pay-for-feedback model. This method is based upon splitting the time interval $T$ into a number of equal length contiguous sub-intervals and randomly sampling a time step in each sub-interval for which it pays for feedback. Using this method on top of our algorithm, we derive the following corollary.

**Corollary 5.** *There exists an oracle efficient online classification algorithm in the pay-for-feedback model for concept classes with small separator sets (size $s$) with groupwise regret upper bounded by* $O\left( T^{1/6}\sqrt{T_{g_i}}(\sqrt{s^{3/2}\log|F_i|} + \sqrt{\log|G|}) \right)$ *(for group $g_i$).*

