# OpenReview forum: "Oracle Efficient Algorithms for Groupwise Regret"
_ICLR.cc/2024/Conference — ICLR 2024 poster_

### Official Review · Reviewer_1CJx · 2023-10-27

**Soundness:** 4 excellent
**Presentation:** 3 good
**Contribution:** 2 fair
**Rating:** 6
**Confidence:** 5

**Summary:**

This work studies an online learning problem in which at each time step a context is obtained that indicates the point belongs to some groups (possibly more than one at the same time). At the end of the game we want to have low regret on all subsequences of points belonging to the same group. The point of this work is that a solution can be build in cases in which the model class is large. A previous algorithm existed that would only handle small model classes. An empirical evaluation is provided.

**Strengths:**

The strength of this paper is obtaining what is claimed in theorem 1 about algorithm 1. Another one is the empirical evidence provided, that checks that the algorithm indeed behaves how it was supposed to do.

**Weaknesses:**

The main weakness of this work is its marginal contribution. Beyond their empirical evidence their contribution is limited to the proof of theorem 2, which is a small step on top of the existing framework of Blum and Lykouris, almost a remark. Still, it could be a publication, I'm giving a weak accept because of this.

**Questions:**

It is not obvious that you should have an "always active" subsequence in order to perform well empirically. It would be good if it is reported that the algorithm does not work so well if this is not present, so others people that want to implement this can take this fact into account.

Theorem 1 and Theorem 3 say essentially the same. It's redundant.

Minor

In the abstract there is "\{1, 2, \cdots T\}" comma missing after \cdots.

"low order regret terms" -> "low-order regret terms"

Thm 1 (informal): contains "of of"

"to each Individual" remove capitalization.

---

> ### Author Response · Authors · 2023-11-16
> **Thanks for the Review**
>
> Thank you for the review and the experimental suggestion. On your suggestion, we have added to the revision new experiments in Appendix F, in which we see that including/excluding the "always on'' expert has a marginal effect on the regret of our algorithm: both are much better than the baseline.
>
> We made the choice to include the "always on'' expert on theoretical grounds. Doing so provides a regret bound over the entire sequence, which does not follow generically from having regret bounds for subsequences. If there are subsequences that partition the entire sequence, then one gets a regret bound on the entire sequence that follows from summing up the regret bounds on the partition but this is wasteful. As an extreme example, consider the case in which each time step corresponds to a separate group. In this case group-wise regret bounds alone do not give any non-trivial regret guarantee over the full sequence. If on the other hand the subsequences do not partition the space, then there is not necessarily a regret bound on the entire sequence without including the "always on'' expert at all. In our experiments, there are indeed (large) groups that partition the sequence, which is consistent with the experimental results showing that the "always on'' expert is not needed in these instances.
>
> We have fixed the minor errors you have pointed out and removed our informal Theorem 1, to help focus the reader on the formal statements of our results.

---

### Official Review · Reviewer_eJaJ · 2023-10-30

**Soundness:** 3 good
**Presentation:** 2 fair
**Contribution:** 4 excellent
**Rating:** 8
**Confidence:** 4

**Summary:**

The paper studies efficient and practical algorithms for group-wise regret-minimization online learning. The problem is a variate of the classical online problem, in which a hypothesis class $H$ is fixed, and a sequence of individuals/features ${x_{1}, \cdots, x_{T}}$ and their labels ${y_1, \cdots, y_T}$ are chosen by an adversary. The algorithm has to make predictions at each time step $t \in [T]$. The classical notion of measuring a ''good’’ algorithm is *regret*, which is defined as the cumulative gap between the cost induced by the online learning algorithm and the cost induced by the best fixed model in the hypothesis class. The group-wise regret minimization problem further requires binding the regret for each group, which, mathematically, can be viewed as a collection of subsequences indexed by the ``mapping’’ of different groups.

The work of BL [ITCS’20] has shown that it is possible to achieve small regret w.r.t. the best model $f_{g}$ for every fixed group $g$ by reducing the problem to the sleeping expert problem. This implies for a large family of online learning problems, there exist algorithms with group-wise $1+o(1)$-multiplicative regret that run in time polynomial of $|G|$ and $|H|$. However, while the size of $G$ is usually small, even some ``elementary’’ models, e.g., linear models, have quite large sizes of $H$, which prevents the algorithm of BL [ITCS’20] from being practical.

The main contribution of the paper is to improve the runtime of the group-wise regret minimization algorithm of BL [ITCS’20], and, in particular, remove the dependence on $|H|$ for the run time. To this end, the paper observes that we can actually solve each policy sub-sequence by external regret algorithms, which require far less time to compute, and treat each ``expert’’ in the overall algorithm as the output of the external regret algorithms. In doing so, the algorithm avoids enumerating over the hypothesis class $|H|$, and only scales w.r.t. $|G|$. The paper then provides some implementations on both synthetic and real-world datasets, and the experimental results of their algorithms are strong.

**Strengths:**

Overall, I like this paper as it provides the practical algorithm for a problem where the existence of the solution (or even, a theoretically-efficient algorithm) has been known, but no practically-efficient algorithm was proposed. This serves as a nice ``bridge’’ between theory and practice. As the paper itself has mentioned, practical group-wise regret-minimization algorithms have various downstream applications, including algorithmic fairness.

I also read through the analysis in Appendix B, and I think they are correct, barring some presentation issues (mentioned in the ``weakness’’ section). Although the technical idea is simple and somehow straightforward to come up with, I do think the result itself is neat and cute. Finally, the experimental results of the paper are quite strong, although the comparison is somehow tailored to your algorithm since the benchmark as they are not designed for group-wise regret minimization.

**Weaknesses:**

One criticism I have when reading the paper is that the paper is not presented in a fully self-contained and rigorous manner. For instance, the proposed algorithm uses AdaNormalHedge as a black box, but the guarantee of such an algorithm is never formally described. (I am aware of the description in Appendix B, but there is no proposition + proper citation for this.) Similarly, when citing external regret minimization algorithms for applications, the formal quantifiers and guarantees for those algorithms are not provided.

Similarly, the introduction is written in a very informal way. I understand this might be a result for the authors to accommodate the broader readership of the conference; however, I think it actually adds to confusion. For instance, when defining the notion of diminishing group-wise regret, it would be much more helpful to include the actual mathematical definition of ''squared error’’ and ''best model in H on that sequence’’. (Also, why the notion is limited to squared error but not general loss functions?)

The same applies to the statement of Theorem 1, in which the notion of ``computationally efficient’’ is not defined(!) The phrase ''best model on hindsight’’ is used in a very informal way – I think you should properly define this notion (overall vs. on group-wise sequences) with the proper quantifiers.

A note for presentation problems in Appendix B: the usage of expectation notation $\mathbb{E}[]$ is rather confusing in this section. Your derivation crucially relies on the control of which coins the expectation is taken upon. I think in this case, the expectation notation should have subscript explicitly stating the source of randomness. Furthermore, the way you talk about $p_{t}^{I} $ vs $z_{t}^{I}$ is not rigorous enough. If I understand it correctly, $p_{t}^{I}$ is a random variable whose supports are some realizations denoted as $z_{t}^{I}$. In light of this, should the term in the first inequality be $\mathbb{E}[\sum_{t} I(t) \ell(p_{t}^{I}, y_{t})]$? Overall, I do think this section has quite some room for improvement.

**Questions:**

Is your notion of computationally efficient in Theorem 1 defined as polynomial time in $T$, $d$, and $n$ (or some other input-related size)? If I understand correctly, what you want to say is that it is reasonable to assume $|G|$ is of polynomial sizes of the input, but $|H|$ is usually quite large. Therefore, your algorithm that does not scale with $|H|$ implies poly-time efficiency.

I don’t quite understand the term ``diminishing/vanishing regret’’ – in your Theorem 2, the term $\sqrt{T_{T} \log(|G|)}$ is not $o(1)$ itself. Are you implicitly enforcing a lower bound on the $\alpha_{I}$?

A MISC comments: The discussion on the technical front by comparing your work with BL [ITCS’20] looks nice. I think you can expand this discussion to give more details, and present it earlier in the paper.

**Details Of Ethics Concerns:**

I do not have any ethics concerns for this work.

---

> ### Author Response · Authors · 2023-11-16
> **Thanks for the Review**
>
> Thank you for the thoughtful review and writing suggestions. We will take this into account as we revise the paper and aim to improve the exposition. In the short term (in the revision we have uploaded) we have added a more formal section on sleeping experts and AdaNormalHedge, as well as a formal statement of its regret guarantees in Appendix B.
>
> We kept the introduction informal for the reason you guessed: to make the goals of the paper accessible to a wider audience --- but we understand your concern that this makes the paper harder to approach for a more technical reader. At the moment, the precise definitions of \'\'no-regret'' and \'\'best-in-hindsight'', are provided in Section 2. We will consider moving them to the introduction. You are right that our results hold for general loss and not just squared loss: we use squared loss as an exemplar both because it is very common, and because our experimental results are for (squared loss) linear regression. We will clarify that our framework is not limited to squared loss.
>
> To answer your questions:  \'\'diminishing external regret'' refers to the regret scaling sub-linearly in the length of the sequence, so that the average regret tends to $0$ with the length of the sequence.  We have added a note to that effect in the introduction. Computationally efficient here is defined as oracle-efficient with respect to the model class $H$ and polynomial run time with respect to the other parameters of the problem ($|G|$, $d$, $n$). This means that our algorithms should run at most a polynomial number of the online learning ``oracles'' and have additional over-head that is polynomial in the problem parameters. We will clarify.
>
> Thanks for pointing out the issue with the first inequality --- there was indeed a typo, which we have fixed now. On your suggestion, we have expanded on the proof of Theorem 2, and in particular have made an effort to be explicit when it comes to the coins over which the expectations are taken.

---

> > ### Comment · Reviewer_eJaJ · 2023-11-20
> >
> > I've read your responses, and I'm satisfied with the answers. I did not carefully check the updated version, but I did see a bunch of flagged problems got fixed with a quick look. (On a side note, I do not think it's a good idea to change the scores based on an updated version *during* the review process. :) )
> >
> > I briefly looked into my colleagues' comments, and I'm not concerned about the correctness of the result as far as I can see. Overall, I'm keeping my score unchanged.

---

> > > ### Author Response · Authors · 2023-11-21
> > >
> > > Thank you! We appreciate the time spent in reviewing. We are especially glad that you are not concerned about correctness, and hope that the reviewing team is in agreement about this.

---

### Official Review · Reviewer_vWf7 · 2023-10-30

**Soundness:** 4 excellent
**Presentation:** 4 excellent
**Contribution:** 3 good
**Rating:** 8
**Confidence:** 3

**Summary:**

The paper studies the problem of online prediction, where at each time step, a new example arrives, and the learner has to make a prediction. The goal is to minimize regret not just overall, but also simultaneously for different subgroups defined based on features. A previous algorithm for this problem (Blum & Lykouris, 2019) provides regret guarantees but is computationally inefficient when the hypothesis class is large. The work proposes a modification to the algorithm of Blum & Lykouris (2019) that reduces the problem to the problem of external regret minimization. The new algorithm uses a significantly smaller number of experts for making the decision at the time step. The algorithm is applied to problems like online linear regression, classification with small separator sets, and linear optimization. Experiments on synthetic and real datasets show substantially lower error and regret compared to standard online learning algorithms.

**Strengths:**

While building on prior work by Blum & Lykouris (2019), the paper introduces a simple yet meaningful modification that enhances computational efficiency. This facilitates the use of large hypothesis classes, such as linear models. The reduction to standard external regret minimization, while expected, remains theoretically novel.

The algorithm's design and analysis are technically sound, and the paper offers a comprehensive set of experiments.

The paper is well-written and easy to follow, with the problem being well-motivated.

Achieving regret guarantees across groups is pivotal. This paper renders it feasible for large model classes, broadening the applicability of these methods.

In summary, this paper presents a theoretically grounded, significant contribution, backed by robust experimental validation.

**Weaknesses:**

I believe that including an experimental comparison with the Blum & Lykouris (2019) approach would better justify the superiority of the new method. Is it possible to conduct such a comparison using a smaller model class?

**Questions:**

See questions.

---

> ### Author Response · Authors · 2023-11-16
> **Thanks for the Review**
>
> Thanks for the suggestion, which we take seriously. We considered whether we should attempt to implement and compare to [BL19], but opted against it. The difficulty with [BL19] is that it requires a small finite class of functions, which is a property that is not realized by any reasonable "hypothesis class''. For example, we focus on linear regression (perhaps the simplest widely used hypothesis class) --- but this is a continuous model class. To run [BL19], it would be necessary to discretize the set of linear predictors: if we discretized each coordinate to granularity $\epsilon$, in $d$ dimensions this would require $(1/\epsilon)^d$ hypotheses. This would be infeasible for anything beyond a very small number of dimensions, and we don't think it would be enlightening. This is  why [BL19] has not been implemented in prior work: indeed, we view the main contribution of our paper as making the approach initiated by [BL19] practical.
>
> References:
>
> [BL19] Avrim Blum and Thodoris Lykouris. Advancing subgroup fairness via sleeping experts. In Innovations in Theoretical Computer Science Conference (ITCS), volume 11, 2020.

---

### Official Review · Reviewer_6fRD · 2023-11-01

**Soundness:** 1 poor
**Presentation:** 2 fair
**Contribution:** 1 poor
**Rating:** 3
**Confidence:** 4

**Summary:**

The paper considers the problem of minimizing the group regret, where the goal is to minimize the regret with respect to each subsequence of trials in the pre-defined group simultaneously. Based on AdaNormalHedge, the authors propose an algorithm for the problem. The experimental results are also shown.

**Strengths:**

The topic is relevant to the machine learning community as it reflects the multi-objective nature of online prediction problems. The experimental results show the proposed method works well in practice. The experimental results show the superiority of the proposed method against baselines.

**Weaknesses:**

I am afraid that the proof of the main theorem (Theorem 2) might be wrong, or at least incomplete. Simply put, the algorithm aggregates sleeping experts where each sleeping expert is awake only when the trial belongs to a designated subsequence of trials. Then, the theorem trivially holds when each subsequence of trials is disjoint to each other, as mentioned in the paper. On the other hand, if subsequences intersect, it is not fully clear if the proof is correct.

**Questions:**

It would be nice if you could comment on my concerns about the proof of Theorem 2.

---

> ### Author Response · Authors · 2023-11-16
> **Thanks for the Review**
>
> Thank you for the review. We are confident that our theorem and proof are correct, and so we would like to understand more specifically what your concern is. We're happy to engage interactively if you have a specific doubt about the proof. Barring that, we attempt here to give an overview of the argument, focusing on intersecting subsequences (which you are correct in noting is the interesting case). We will here give a high level overview, because the structure of the argument is quite simple. To start, it is important to understand the sleeping experts setting.
>
> The **Sleeping experts** setting (introduced in [FSSW97]): At the start of round $t$, each of the experts may be "**awake**"(i.e., available to make a prediction) or "**asleep**’’(not available to make a prediction). Before the algorithm makes a prediction at round $t$, it learns which of the experts are awake at round $t$, and also knows the loss incurred by the experts (and their awake/asleep status) in all previous rounds. The algorithms in [FSSW97], [BM07], [LS15] use the history to compute the probability with which to pick amongst the "awake’’ experts at round $t$, and follow their prediction. Note crucially that multiple experts can be awake simultaneously.
>
>
> For completeness we have added a description of the AdaNormalHedge algorithm [LS15] in Appendix B. The theorem proven by [LS15] guarantees that  AdaNormalHedge plays distributions over experts such that simultaneously for each expert, when restricted to the subsequence of rounds for which that expert is awake, the algorithm's regret to that expert is  is $O(\sqrt{T_I \log{|\mathcal{I}|}})$, where $T_I$ is the number of rounds for which expert $I$ is awake.   See Eq (3) in Appendix B. Crucially, this guarantee holds simultaneously for each expert *even though the subsequences on which the experts are awake are intersecting*.
>
> Our reduction then instantiates a sleeping experts algorithm (AdaNormalHedge) in which the experts are themselves no-regret learning algorithms, with one "assigned'' to each group. The learning algorithm assigned to a group $g$ is "awake'' on those rounds for which the example is a member of group $g$. Moreover, we provide feedback to and update the state of the no-regret algorithm assigned to group $g$ only on those rounds for which the example is a member of group $g$. Since the individual no-regret algorithms have regret guarantees for adversarially chosen sequences (including the sequence we choose to feed it), each no-regret algorithm provides a sequence of suggestions that would obtain cumulative loss comparable to that of the best fixed hypothesis on the subsequence on which it is run --- which is just the subsequence of examples belonging to a particular group. These groups intersect --- which is ok, because there is nothing stopping us from updating the algorithms in parallel. Of course we can only adopt the action suggested by a single one of the algorithms at each round --- but this aggregation is done by AdaNormalHedge, which is able to choose actions in such a way so that its regret on each subsequence corresponding to a group is not much larger than the regret of the expert assigned to that group --- which is in turn not much larger than the best hypothesis on that group. In the end, the two regret guarantees (the regret guarantee for AdaNormalHedge and the regret guarantee for the no-regret-learning algorithm assigned to the group) simply add up. We have revised the draft in an attempt to add clarity to the proof. Once again, if you have specific concerns we're happy to engage with them.
>
> References:
>
> [BL20] Avrim Blum and Thodoris Lykouris. Advancing subgroup fairness via sleeping experts. In
> Innovations in Theoretical Computer Science Conference (ITCS), volume 11, 2020.
>
> [BM07] Avrim Blum and Yishay Mansour. From external to internal regret. Journal of Machine Learning
> Research, 8(6), 2007.
>
> [FSSW97] Yoav Freund, Robert E. Schapire, Yoram Singer, and Manfred K. Warmuth. Using and combining
> predictors that specialize. In Proceedings of the Twenty-Ninth Annual ACM Symposium on Theory
> of Computing, STOC ’97, page 334–343, New York, NY, USA, 1997. Association for Computing
> Machinery.
>
> [LS15] Haipeng Luo and Robert E. Schapire. Achieving all with no parameters: Adaptive normalhedge,
> 2015

---

### Meta-Review · Area_Chair_ntUs · 2023-12-06

**Metareview:**

The paper studies online prediction when there are multiple groups and we would like to get good regret simultaneously for all groups. There are previous works for the problem but they are intractable for large model classes because they reduce to sleep experts where each expert is a pair of a group and a model for that group. Instead, the present work uses one expert for each group and that expert utilizes an oracle for finding the best model for that group. For many large classes of model, such an oracle exists and the result goes through. All reviewers agree that the empirical results are strong, showing good runtime for several settings.

On the other hand, the reviewers have diverging opinions on the proof. Two reviewers suggest that the proofs are simple, building on top of existing machinery (the entire proof is about 1 page long). One reviewer suggests that the proof is incomplete though they have not engaged with the authors and other reviewers to clarify on the problem.

**Justification For Why Not Higher Score:**

The proof is a simple observation on top of existing results.

**Justification For Why Not Lower Score:**

The empirical results are strong, showing fast runtime in several settings where efficient algorithms were not known.

---

### Decision · Program_Chairs · 2024-01-16

Accept (poster)